# Application of an Enhanced Whale Optimization Algorithm on Coverage Optimization of Sensor

**DOI:** 10.3390/biomimetics8040354

**Published:** 2023-08-09

**Authors:** Yong Xu, Baicheng Zhang, Yi Zhang

**Affiliations:** College of Electrical and Computer Science, Jilin Jianzhu University, Changchun 130119, China; xuyong@jlju.edu.cn (Y.X.); zhangbc579@163.com (B.Z.)

**Keywords:** whale optimization algorithm, Lévy flight, distributed generation, wireless sensor network

## Abstract

The wireless sensor network (WSN) is an essential technology of the Internet of Things (IoT) but has the problem of low coverage due to the uneven distribution of sensor nodes. This paper proposes a novel enhanced whale optimization algorithm (WOA), incorporating Lévy flight and a genetic algorithm optimization mechanism (WOA-LFGA). The Lévy flight technique bolsters the global search ability and convergence speed of the WOA, while the genetic optimization mechanism enhances its local search and random search capabilities. WOA-LFGA is tested with 29 mathematical optimization problems and a WSN coverage optimization model. Simulation results demonstrate that the improved algorithm is highly competitive compared with mainstream algorithms. Moreover, the practicality and the effectiveness of the improved algorithm in optimizing wireless sensor network coverage are confirmed.

## 1. Introduction

The rapid development of Internet of Things (IoT) technology has significantly improved people’s lives and productivity [1,2] in recent years. Fifth-generation mobile communication technology (5G) advancement will further propel societal development [3]. Wireless sensor networks (WSN) consist of sensor nodes with sensing and communication capabilities and are fundamental components of the IoT [4]. These nodes can perceive, process, and transmit information within a target area, enabling monitoring across various terminals and transforming human interaction with nature. Consequently, WSN research has garnered increasing attention. WSNs have substantial research and application value in diverse fields, such as healthcare, environmental protection, meteorological monitoring, and military defense [5], and have profoundly impacted global technological progress. To effectively monitor a target area, WSNs must provide sufficient coverage. The optimal deployment of sensor nodes to cover a larger area with fewer nodes has become a research hotspot in WSN optimization [6].

Many scholars have used various methods to address the coverage problem in wireless sensor networks. Yoon Y. and Kim Y. H. [7] derived the upper and lower bounds on the coverage of a 2D deployment of static sensors. They used these bounds to construct a method of estimating the coverage of a deployment by assuming that there are only pairwise intersections between the disks representing the range of each sensor. This MA outperforms the previous techniques regarding both speed and coverage achieved. Liu, C and Du H [8] propose an algorithm named 2-partition sweep coverage (2-PSC) based on a partition of the coverage time requirements and positions to achieve t using a K-sweep coverage with the minimum number of mobile sensor nodes, where t is the sweep period constraint to complete the entire coverage process and K is the set of coverage time requirements. Wang W.M. et al. [9] proposed a k-equivalent radius enhanced virtual force algorithm (k-ERVFA) to optimize uneven regional coverage for different k-coverage requirements. Theoretical analysis and simulation experiments are carried out to demonstrate the effectiveness of our proposed algorithm. Paulswamy S.L. et al. [10] proposed a new disc shape deployment strategy. The proposed deployment strategy provides desirable coverage and requires an increased number of sensor nodes when compared with the hexagon shape deployment strategy. The authors employed different methods to achieve a network coverage of sensors, and with the rise of artificial intelligence, more viable solutions have been proposed for this type of problem.

The wireless sensor coverage optimization problem is solvable with optimization strategies. Recently, numerous researchers have begun to propose different swarm intelligent optimization algorithms (SIA) to tackle such problems. SIAs are meta-heuristic algorithms that simulate the behavior of animal groups such as fish, birds, bees, and wolves, optimizing outcomes through simple, limited interactions between individuals and information exchange and cooperation within groups. Priyadarshi, R and Gupta, B [11] introduced an improved particle swarm optimization (PSO) algorithm to optimize coverage with minimal nodes. Zhu, WB et al. [12] suggested a dual-tuned simplified group optimization (SSO) algorithm to maximize coverage areas and improve WSN performance. Nematzadeh S et al. [13] presented a mutant GWO (MuGWO) to enhance resource utilization by maximizing coverage and maintaining connectivity. Dao, TK et al. [14] proposed an improved Archimedes optimization algorithm (EAOA) to address optimal node coverage in unbalanced WSN distribution during random deployment. ZainEldin, H et al. [15] developed an improved dynamic deployment technique based on a genetic algorithm (IDDT-GA) to maximize coverage with minimal nodes and reduce overlapping areas between adjacent nodes. Although these intelligent optimization algorithms have somewhat improved target area coverage in WSNs, they have limitations, such as low search accuracy and susceptibility to locally optimal solutions in PSO and GWO, high time complexity in ALO, and sensitivity to parameter settings in AOA. The whale optimization algorithm (WOA) is a meta-heuristic optimization algorithm that simulates humpback whale hunting behavior, and was proposed by Mirjalili, S and Lewis, A. D [16]. Compared with other commonly used swarm intelligent optimization algorithms, WOA uses random or optimal search agents to simulate whale hunting behavior and a spiral mechanism to mimic the humpback whale’s bubble net attack method. The algorithm has a simple mechanism, few parameters, and strong optimization capabilities, widely recognized in the industry. However, WOA still requires improvement when solving optimization problems, driving researchers to study it further and to propose various enhancement strategies for it. Zhang, J and Wang, J.S. [17] introduced an improved WOA based on nonlinear adaptive weight and golden sine operator (NGS-WOA) to enable search agents to adaptively explore the search space and balance development and exploration phases. Liu, J.X. et al. [18] proposed an enhanced global exploration WOA (EGE-WOA) to improve convergence behavior and global exploration efficiency. Kaur, G and Arora, S [19] incorporated chaos theory into WOA to improve global convergence speed and performance. Bozorgi, SM and Yazdani, S [20] combined WOA’s development with DE’s exploration to offer a promising candidate solution. Luo, J, and Shi, BY [21] proposed a hybrid WOA called MDE-WOA, embedding an improved differential evolution operator to accelerate convergence and improve accuracy.

Mafarja M. et al. [22] introduced SWOA and VWOA and used them as search strategies in a wrapper feature selection model. They tested the algorithms on nine different high-dimensional medical datasets, with a low number of samples and multiple classes. Their results reveal superior performance of the VWOA over the SWOA and other approaches used for comparison purposes. Zhang M.L. et al. [23] proposed an efficient, intelligent prediction model based on the machine learning approach, which combines the improved whale optimization algorithm (RRWOA) with the k-nearest neighbor (KNN) classifier to offer early identification and intervention of critical illnesses in patients. The model offers a scientific framework to support clinical diagnostic decision making. Shivahare B.D. and Gupta S.K. [24] addressed automated segmentation and classification of COVID-19 and normal chest CT scan images. They introduced a variant of the whale optimization algorithm named the improved whale optimization algorithm (IWOA). The IWOA is efficient and achieved better segmentation evaluation measures and better segmentation masks than other methods. It can detect COVID-19 disease from chest CT scan images within a shorter period and can help doctors to start COVID-19 treatment at the earliest.

In the process of improving the WOA algorithm, many scholars have also incorporated ideas from other intelligent optimization algorithms and combined one or more different algorithms with WOA to form new algorithms. Tong W.Y. [25] embedded the DE/rand/1 operator of differential evolution (DE) and the mutation operator of the backtracking search optimization algorithm (BSA) into WOA to form two new algorithms under the proposed framework, called WOA-DE and WOA-BSA. WOA-DE and WOA-BSA are competitive compared with some state-of-the-art algorithms. Prabhakar D. and Satyanarayana M. [26] combined salp swarm optimization (SSA) and whale optimization algorithm (WOA) to propose a new algorithm called salp swarm whale optimization algorithm (SSWOA). In this new algorithm, the SSA algorithm guides the evolution and the WOA algorithm provides assistance. This new algorithm exhibits high convergence accuracy and fast convergence speed. Mohammed H. and Rashid T. [27] have proposed a new algorithm called WOAGWO based on the whale optimization algorithm (WOA) and grey wolf optimization (GWO). They embedded GWO’s hunting mechanism into the development phase of WOA and added a new technique in the exploration phase to improve the solution after each iteration. Their experimental results also confirm that the algorithm performs well and can achieve optimal solutions. All of the above algorithms have shown good optimization performance on existing problems.

This paper presents a novel enhanced WOA (WOA-LFGA) based on the Lévy flight and genetic algorithm optimization problem mechanism applied to the coverage optimization problem of wireless sensor networks.

The remainder of this paper is organized as follows: The remaining part of this section covers the traditional WOA concept and mathematical model, the basic principles of Lévy flight, the genetic algorithm’s crossover and mutation processes, and the wireless sensor coverage optimization model; Section 2 presents the basic framework of the new enhanced WOA (WOA-LFGA) based on Lévy flight and genetic algorithm optimization problem mechanism; Section 3 tests the improved algorithm using 29 standard test functions and applies it to the wireless sensor coverage optimization problem, comparing the proposed algorithm with other mainstream swarm intelligent optimization algorithms; Section 4 concludes the paper and proposes future work.

The main contributions of this paper include:

1. The proposal of an improved WOA called WOA-LFGA, based on Lévy flight and genetic algorithm optimization problem mechanism, significantly enhancing the global optimization ability and convergence accuracy of the algorithm.

2. The introduction of a WSN coverage optimization method based on WOA-LFGA. Simulation results demonstrate that, compared with other mainstream algorithms, the proposed algorithm exhibits strong competitiveness, further validating the practicability and effectiveness of WOA-LFGA in optimizing wireless sensor network coverage.

### 1.1. Wireless Sensor Network Coverage Model

Assume that *m* sensor nodes are deployed in a two-dimensional monitoring area S = {*s*_1_, *s*_2_,…, *s_m_*}, where the coordinate of *s_i_* is denoted by (*x_i_*, *y_i_*) and that *i* = 1, 2, …, *m*. This paper adopts the Boolean model as the node perception model, and the monitoring area is discretized into a rectangle with *L × W* pixels. The probability of the monitoring point *tj* being perceived by node *s_i_* is:(1)$$p\left(s_{i},t_{j}\right)=\left\{\begin{array}{c} 1\;\;\;\;if\;d\left(s_{i},t_{j}\right)\leq r_{s}\\ 0\;\;\;\;otherwise\;\;\;\;\;\;\;\; \end{array}\right.$$
where, *r_s_* is the sensing radius of the sensor, and $d\left(s_{i},t_{j}\right)$ is the Euclidean distance between the sensor node and the monitored node, expressed as:(2)$$d\left(s_{i},t_{j}\right)=\sqrt{{\left(x_{i}-x_{j}\right)}^{2}+{\left(y_{i}-y_{j}\right)}^{2}}$$

Then the probability of *t_j_* being covered in WSN is:(3)$$P\left(S,t_{j}\right)=1-\prod_{i=1}^{m}\left[1-p\left(s_{i},t_{j}\right)\right]$$
where, *S* is all wireless sensor nodes in the region. Assuming that the monitoring area is equivalent to *L × W* pixel points, and the coverage rate of the sensor deployment area can be defined as:(4)$$f=\frac{\sum_{i=1}^{L}\sum_{j=1}^{W}P\left(S,t_{\left(i-1\right)W+j}\right)}{L\times W}$$

To further evaluate the algorithm’s performance, we introduce a coverage efficiency metric *C*, which is defined as the ratio of the total coverage area of all nodes to the total sensing area of all sensor nodes in a wireless sensor network. Formula (5) describes its definition.
(5)$$C=\frac{f\times L\times W}{N\times \pi \times r_{s}^{2}}$$

The coverage efficiency metric *C* quantitatively reflects the redundancy of deployed sensor nodes, where a higher value of *C* indicates a lower redundancy of nodes and a more even distribution of nodes, while a lower value of *C* indicates a higher redundancy of nodes and more node clustering.

In this optimization model, our objective is to maximize *f* and *C* by altering the positions of the wireless sensors, denoted as *S*. The range of *S* is constrained by the region size, which means that the wireless sensor nodes must move within a specified space.

### 1.2. Overview of Whale Optimization Algorithm (WOA)

In the whale optimization algorithm, the position of each whale represents a feasible solution to the problem. During whale hunting, each humpback whale’s hunting methods fall into two categories: encircling the prey and using a bubble net attack, spiraling up to repel and encircle the prey. During each iteration, the whales randomly choose to prey with one of these two behaviors. The algorithm generates a random number *p*, in the range of [0, 1]. When *p* < 0.5, the whale performs the encircling behavior as described by Formulas (7) or (9). When *p* ≥ 0.5, the whale attacks the prey using bubble net as described by Formula (12). In the process of encircled humpback whale hunting, whales will choose to move towards the best-known individual whale in the current population when |A| < 1, which can be described by Formulas (6) and (7), or randomly select a whale and move in its direction when |A| ≥ 1, which can be described by Formulas (8) and (9). In the following formula, we define $\vec{X}\;\left(t\right)$ as the position vector of the whale in the current iteration, and $\vec{X}\;\left(t+1\right)$ as the new position vector of the whale in the next iteration.
(6)$$\vec{D_{1}}=\left|\vec{C}\cdot \vec{X^{*}}\;\left(t\right)-\vec{X}\;\left(t\right)\right|$$
(7)$$\vec{X}\;\left(t+1\right)=\vec{X^{*}}\left(t\right)-\vec{A}\cdot \vec{D_{1}}$$
(8)$$\vec{D_{2}}=\left|\vec{C}\cdot \vec{X_{rand}}-\vec{X}\;\left(t\right)\right|$$
(9)$$\vec{X}\;\left(t+1\right)=\vec{X_{rand}}-\vec{A}\cdot \vec{D_{2}}$$
where *t* is the current iteration number; the dot notation “·” is an element-by-element multiplication; | | is the absolute value; $\vec{A}$ and $\vec{C}$ are coefficient vectors; $\vec{X^{*}}$ is used to obtain the current position of the optimal individual whales; $\vec{X_{rand}}$ is used throughout the whale populations to obtain randomly selected individual whale position vectors. The coefficient vector $\vec{A}$ and $\vec{C}$ computation formula is as follows:(10)$$\vec{A}=2\vec{a}\cdot \vec{r}-\vec{a}$$
(11)$$\;\vec{C}=2\vec{r}$$

Including $\vec{a}$ in an iterative process, linear cut from 2 to 0; $\vec{r}$ is the range of random vectors between [0, 1].

Bubble net hunting is another method by which humpback whales hunt. While using bubble net to drive away prey, whales will constantly update their position. This process stimulates the spiral attack of the whales, and the formula is as follows:(12)$$\vec{X}\;\left(t+1\right)=\vec{D^{\prime }}\cdot e^{bl}\cdot \cos\left(2\pi l\right)+\vec{X^{*}}\;\left(t\right)$$
(13)$$\vec{D^{\prime }}=\left|\vec{X^{*}}\;\left(t\right)-\vec{X}\;\left(t\right)\right|$$
where *b* is the constant used to define the shape of the logarithmic spiral and *l* is a random number in the range [−1, 1].

### 1.3. The Lévy Flight Method

Lévy flight has been widely used in various optimization algorithms, and the results show that it can provide good global search capability for algorithms. The Lévy flight method not only ensures the diversity of the population but also improves the convergence speed and accuracy of the algorithm. During the flight, short-distance movements with smaller steps and long-distance movements with larger steps are carried out alternately. This is conducive to increasing the diversity of the population and avoiding the algorithm falling into a local optimal solution. In this paper, we integrate the Lévy flight method [28] into the exploration phase of WOA, the formula improved by Lévy flight can be expressed as:(14)$$\vec{X}\;\left(t+1\right)=\left\{\begin{array}{l} sign\left(rand-\frac{1}{2}\right)\cdot \alpha \cdot \left[\vec{X_{rand}}-\vec{X}\;\left(t\right)\right]⊕Lévy\left(s\right)\;\;\;\;if\;p_{2}>0.95\\ \vec{X^{*}}\;\left(t\right)+\vec{F}\cdot \alpha \cdot \left[\vec{X^{*}}\;\left(t\right)-\vec{X}\;\left(t\right)\right]⊕Lévy\left(s\right)\;\;\;\;\;\;\;\;\;\;\;\;\;\;\;\;otherwise \end{array}\right.$$
where *p*_2_ is a random number within the range of [0, 1], Lévy flight is a non-Gaussian random process with smooth increments obeying Lévy stable distribution, and its formula is expressed as (15).
(15)$$Lévy\;\left(s\right)∼{\left|s\right|}^{-1-\beta },\;\;\;\;0<\beta \leq 2$$
where *s* refers to the Lévy flight of step length and *β* the index, which we will assign to 1.5. *s* can be calculated by the formula as follows:(16)s =uv1β,    u~N0,σu,    v~N0,σv
where, *u* and *v* are subject to normal distributions, enabling individuals to obtain effective positioning in the search space and thus enhancing the algorithm’s exploration ability.$\;\sigma_{u}\;$ and $\;\sigma_{v}\;$ are expressed in the following formula:(17)σu=Γ1+β·sinπβ2β·Γ1+β/2·2β−1/21β
(18)$$\sigma_{v}=1$$
where, Γ is the standard gamma function.

In this study, to enhance the global exploration capability of the improved algorithm, we replaced Formula (8) in the original algorithm with Formula (14). In other words, when the conditions *p* < 0.5 and |A| ≥ 1 are satisfied, the algorithm employs Equation (14) to update the position of the whales. This equation is specifically utilized to perform position updates using a Lévy flight approach, which enhances the algorithm’s global exploration capabilities.

### 1.4. Genetic Algorithm

The genetic algorithm is a swarm intelligent optimization algorithm based on Darwinian evolution. Its main idea simulates the natural selection law of survival of the fittest. In the genetic algorithm, each solution is encoded as a chromosome, and the fitness function in the optimization algorithm calculates the adaptability of each chromosome to the living environment. The better the fitness value is, the stronger the adaptability of the individual to the environment, and the higher the probability of being retained in nature; conversely, the worse the fitness value is, the weaker the adaptability of the individual to the environment, and the easier it is to be eliminated in the process of iteration. The genetic algorithm evolves the optimal solution of the problem through N generations of heredity, variation, crossover, and replication. Crossover and mutation are at the heart of the algorithm.

Mutation refers to randomly replacing values on a chromosome with other values and comparing the mutated chromosome with the original, keeping the one with better fitness. The process can be clearly shown in Figure 1.

## 2. Proposed WOA-LFGA

This section introduces the details of WOA-LFGA, an improved algorithm based on WOA. The improvement of WOA in this paper includes the initialization phase, development phase, genetic optimization mechanism, and boundary processing strategy. The mathematical model and pseudo code of WOA-LFGA are presented.

### 2.1. Initialization Based on Chaotic Map

This section introduces the details of an improved algorithm WOA-LFGA based on WOA. Although WOA has a good convergence rate, it still cannot perform well in the global search process. Therefore, to ensure that individual whales have strong searching abilities at the beginning, this paper introduces chaotic mapping to initialize the population. Chaotic mapping has randomness, ergodicity, and initial value sensitivity, which can make the algorithm converge faster. In [19], 10 different chaotic maps are described. After conducting multiple experiments, we ultimately selected tent mapping to generate chaotic sequences and initialize the population, so that the initial solutions are distributed as evenly as possible in the solution space. This paper’s improvements to WOA include the initialization phase, development phase, genetic optimization mechanism, and boundary processing strategy. The mathematical model and pseudo code of WOA-LFGA are presented.
(19)$$\vec{X_{t+1}^{k}}=\left\{\begin{array}{c} \frac{X_{t}^{k}}{u},\;\;\;\;\;\;0\leq X_{t}^{k}-lb\leq \left(ub-lb\right)*u\\ \frac{1-X_{t}^{k}}{1-u},\;\;\;\;\;\;\;\;\;\;\;\;\;\;\;\;\;\;\;\;\;\;\;\;\;\;\;\;\;\;\;\;\;\;\;\;otherwise \end{array}\right.$$
where *k* is the population dimension; *t* is the number of current iterations; *lb* is the lower boundary value of the search space; and *ub* is the upper boundary value of the search space. To maintain the randomness of the initialization information of the algorithm, the value of *u* in this algorithm is, after many experiments, 0.3.

### 2.2. Enhanced Exploitation Phase

The Lévy flight method can provide the algorithm with a good global search capability. In this paper, we integrate the Lévy flight method into the exploration phase of WOA, so that individual whales can have a relatively high probability of taking long strides in the iterative process, to expand the search scope and to improve the global search capability of the algorithm. To a certain extent, the introduction of the Lévy flight can also accelerate the cover algorithm’s convergence rate. The iterative formula improved by Lévy flight can be expressed as Equation (14).

Where *t* is the number of current iterations; $\vec{X_{rand}}$ is used throughout the whale populations to obtain randomly selected individual whale position vectors; rand and p2 are random numbers in the range [0, 1]; $\vec{F}$ is the length and $\vec{X}\;\left(t\right)$ is the same random vector (t), in the range [2, 2]; The specific mathematical model of Lévy(s) has been introduced in the previous section; *α* is step size parameter, which can be expressed in the following formula:(20)$$\alpha \;=rand(1,\;dim)*\alpha_{0}$$
where the value of $\alpha_{0}$ is 1.6, the *rand* (1, *dim*) ranges in a random number between 1 and whale individual dimension value.

### 2.3. An Improved Method Based on Genetic Algorithm

The genetic algorithm simulates the process of natural selection, and its core is crossover and variation. We integrate the ideas of crossover and mutation in genetic algorithm into WOA. Crossover can improve the local optimization ability of the algorithm, and mutation can improve the random search ability of the algorithm. At the end of each iteration, the algorithm will select the top 10% of individuals with the best fitness in the population as elite individuals and the bottom 20% of individuals with the worst fitness as elimination individuals. Through crossover and mutation strategies, new chromosome vectors are generated to replace the chromosome vectors of eliminated individuals. In the selected individuals, the variation rate was 0.2.

The crossover process involves randomly selecting two chromosomes from elite individuals’ chromosomes, one for the father and one for the mother. The two chromosomes are then cut off at one point and spliced together to create a new chromosome. This new chromosome contains both a certain amount of the father’s genes and a certain amount of the mother’s genes. The process of mutation uses Formula (21) to update the new position of the eliminated individual:(21)$$\vec{X}\;\left(t+1\right)=\left(ub-lb\right)*\exp\left(\frac{t}{maxiter}\right)+lb$$
where *t* is the number of current iterations; *maxiter* is the total number of iterations; *lb* is the lower boundary value of the search space; and *ub* is the upper boundary value of the search space. The curve of Equation (20) can be visualized in Figure 2.

Through many experiments, we found that, for the optimization of a problem, it is best to introduce a genetic algorithm optimization mechanism when the number of iterations of that algorithm reaches 20% of the maximum number of iterations.

### 2.4. Boundary Processing Strategy

When the individual whale exceeds the boundary, which strategy to employ to pull the individual whale back into the search space becomes a problem that all variation strategies must deal with. The processing strategy of the original WOA is to place the offending whale individuals on the boundary or the multiple of the boundary, which causes a problem. After the completion of an iteration, many whale individuals will be reset on the boundary, and the number of whale individuals in the search space will be reduced. Equation (22) is used in this paper to deal with individual whales that cross the boundary. This strategy will ensure that the entire whale population is randomly distributed in the space, increasing the utilization rate of the entire whale population.
(22)$$\vec{X}\;\left(t+1\right)=\left(\vec{X}\;\left(t\right)-lb\right)\;\%\;\left(ub-lb\right)+lb$$
where *t* is the number of current iterations; *lb* is the lower boundary value of the search space; *ub* is the upper boundary of the search space and *%* is the mod operator.

The pseudocode of the improved algorithm WOA-LFGA can be described by Algorithm 1.
**Algorithm 1:** WOA-LFGAInput: Fitness functionOutput: Available optimal solution(i) Initialization processStep1: Initialize parameter and variable values used in the algorithm.Step2: Initialize the whales population X = X_i_ (i = 1, 2,…, N) using chaotic mapping by Equation (19).Step3: Calculate the fitness for X and select the best individual and assign it to X*.Step4: Set the iteration counter to t = 0.(ii) Iterative processStep5: While t < maxiter, Do.Step6: Update the position for X_i_ by Equation (7) (if *p* < 0.5 and |A| < 1) or Equation (14) (if *p* < 0.5 and |A| ≥ 1) or Equation (12) (if *p* ≥ 0.5).Step7: Select the best 10% and the worst 20% of individuals and use crossover and mutation strategies to update individuals for the worst 20% based on the best 10% of individuals.Step8: Return the search agents that go beyond the boundaries of the search space using Equation (22).Step9: Calculate the fitness for X and update X* if there is a better solution.Step10: Iterate the counter t = t + 1.End.(iii) Results obtainedStep11: Output the best agent X*.The end.

## 3. Results and Discussion

In this section, we use MATLAB R2016 to conduct simulation experiments. The algorithm runs on Windows 10 64-bit system with 8GB memory. The improved algorithm is tested with 29 standard test functions and applied to the wireless sensor coverage optimization problem. The improved algorithm proposed in this paper is compared with several other mainstream swarm intelligent optimization algorithms.

### 3.1. WOA-LFGA for Function Optimization

In this section, the numerical efficiency of the WOA-LFGA algorithm that is improved in this paper is verified by solving 35 mathematical optimization problems. The 35 reference functions can be divided into three categories. Among these, F1–F10 is a single-mode reference function, which reflects the exploration performance of the algorithm. F11–F29 is a multi-modal reference function, which challenges the exploration capability of the algorithm and reflects the development capability of the algorithm. F30–F35 is a composite reference function proposed in CEC 2005. These reference functions are shift, rotation, expansion, and combination variables of some mathematical optimization problems, which are used to test the global optimization ability of the algorithm. These functions can reflect the ability of the algorithm to escape from the local optimal. We compare the WOA-LFGA algorithm with several other recently proposed population intelligent optimization algorithms. The search range space and optimal value f min of test functions and individuals are listed in Table 1, Table 2 and Table 3.

For each reference function, the number of iterations of the algorithm is set to 500 and the population size is 30. The program is repeated 30 times, and its mean and variance are calculated. We compared WOA-LFGA with PSO [29], AOA [30], GWO [31], SSA [32], and WOA, and reported the statistical results in Table 4 and Table 5.

The functions F1–F10 are single-mode reference functions with only one global optimal value in the search space. They are used to evaluate the development capability of the studied meta-heuristic algorithm. As can be seen from Table 4, WOA-LFGA has strong competitiveness compared with other meta-heuristic algorithms, especially the most effective optimization effect in functions F1–F5 and F8–F10, and the optimization effect in F6 and F7 are also more robust than most optimization algorithms. Therefore, WOA-LFGA has excellent exploration performance and local optimization ability.

Functions F11–F29 are multi-modal reference functions. Different from single-modal functions, multi-modal functions contain many locally optimal solutions, and the number of optimal local values increases exponentially with the increase of function dimension. Therefore, these functions are well suited to test the development capabilities of algorithms. As can be seen from Table 4, WOA-LFGA has excellent development capability in both multi-modal reference functions and fixed dimensional multi-modal reference functions. In most test problems, this algorithm is always either the most efficient, sub-optimal, or close to the optimal value. It has been proved that WOA-LFGA has good global search ability and can jump out of optimal local ability.

Functions F30–F35 are some composite reference functions, and optimizing such functions is challenging. According to the optimization data in Table 5, WOA-LFGA achieved the best fitness in three test problems and had strong competitiveness in the results of the remaining three tests. This proves that WOA-LFGA has strong global optimization ability.

The comparison of convergence curves between WOA-LFGA and other population intelligent algorithms is shown in Figure 3. To establish a more intuitive picture, a semi-logarithmic graph is used in this paper to reflect the decline rate of the fitness curve, meaning that the part not shown in the graph indicates the way in which its fitness value has declined to infinitesimal.

As seen in the figure, WOA-LFGA exhibits three distinct convergence trends during the iterative process. In some cases, the algorithm converges to the optimal global solution in less than half of the iterations. This is attributed to the introduction of the Lévy flight method, which enhances the global search scope and convergence speed of the algorithm. As a result, WOA with Lévy flight can locate the optimal global solution more quickly than other algorithms. This behavior is evident in F1, F3, F8, and F14. In other instances, the convergence rate accelerates when the algorithm is near 20% of the total iterations. This phenomenon results from the incorporation of the genetic algorithm optimization problem mechanism. The crossover strategy strengthens the algorithm’s local optimization ability near the optimal individual, while the mutation strategy increases the population’s diversity. This behavior is apparent in F4, F7, F9, and F28. Finally, rapid convergence in the initial steps of iteration is achieved due to the introduction of chaotic mapping for initialization. This approach allows whale individuals to distribute and search the space more evenly during initialization, thereby finding the optimal solution more quickly. This behavior is evident in F5, F11, F19, and F21. In summary, the results of this section show that the improved WOA’s global search and local optimization abilities have been significantly enhanced. Specifically, compared with other population-based intelligent algorithms, the fitness of the proposed algorithm decreases rapidly within fewer iterations and reaches the lowest fitness result more quickly.

To test the scalability of WOA-LFGA, we conducted experiments on 19 benchmark functions (F1–F19) in 4 dimensions (30, 50, 100, 500) and compared them with several metaheuristic algorithms as shown in Table 6. For each algorithm, the population size and maximum number of iterations were fixed at 30 and 500, respectively, and we ran the experiments independently 30 times.

These results indicate that WOA-LFGA outperforms the compared algorithms in most cases, as it achieves the best average value in 53 out of 57 cases (92.98%). This is higher than other algorithms such as AOA (17.54%) and WOA (8.77%), among others (0%). WOA-LFGA is competitive in searchability and convergence rate, demonstrating universality, robustness, and high stability. In the following chapters, WOA-LFGA will be tested in the application of more challenging wireless sensor coverage optimization problems.

### 3.2. WOA-LFGA for WSN Coverage Optimization Problem

To verify the effectiveness of WOA-LFGA in the WSN coverage optimization problem, we compare the proposed algorithm with several other population intelligent algorithms and several other improved WOAs. Equations (4) and (5) are together used as the objective function when solving the WSN coverage optimization model. During the experiment, we found that WOA-LFGA was unable to manage its optimal performance in the coverage optimization problem of wireless sensors. After several experiments, we updated Formulas (21)–(23) for the coverage optimization problem of a wireless sensor:(23)$${\vec{X}}_{j}^{t+1}=\vec{cand}\left[ind\right]$$
where $\vec{cand}$ is a vector subject to rectangular distribution, its range is between (*lb*, *ub* + *r*), its dimension is (*ub* − *lb* + *r*)/*r*; *r* is the node radius of the wireless sensor; *ind* is the index coefficient, and its calculation formula is expressed as Formula (24):(24)$$ind=\left\{\begin{array}{l} j\%r+1\;\;\;\;\;\;\;\;\;\;\;\;\;\;\;\;\;\;\;\;\;\;\;\;\;\;\;\;\;\;\;\;\;\;\;\;\;\;\;\;\;\;\;\;\;\;\;\;\;\;\;\;\;\;\;\;\;\;\;\;\;\;\;\;\;\;if\;j\;is\;odd\\ \lfloor -cos\frac{\pi j}{2}*\left(\lceil \frac{j}{r}\rceil \%2\right)+\left|1+\left|cos\frac{\pi j}{4}\right|+\lfloor \frac{2j-2}{r}\rfloor \right|\rfloor \;\;\;\;if\;j\;is\;even \end{array}\right.$$
where *j* is the dimension index of the whale individual, and % is the mod operator.

The aim of this experiment is to test and compare the improved WOA algorithm with the above five algorithms. The evaluation index is the average overall coverage rate and algorithm stability, which is represented by variance. We tested the coverage of 27 target points within an area of 100 m × 100 m, in which the coverage radius of each target point was 11 m. To make the experimental conclusions more persuasive, we conducted 30 experiments, with 200 iterations for each experiment. The parameter settings of the experiment are shown in Table 7.

#### 3.2.1. Comparison of WOA-LFGA with Other Basic Algorithms

In this section, the effectiveness of WOA-LFGA is measured by comparing it with SMA [33], DOA [34], AOA, BWO [35], and WOA. The parameter settings of the comparison algorithm are all taken from the corresponding literature. In this experiment, the algorithm proposed in this paper is compared with the above five population intelligent algorithms, and the program operation results are shown in Table 8 and Figure 4.

It can be seen from Table 8 that, compared with SMA, DOA, AOA, BWO, and WOA, WOA-LFGA has significantly improved the coverage optimization of WSN. Overall, the optimization effect of SMA, AOA, and BWO in this experiment could be better, and the average coverage rate is below 70%. The other three algorithms have relatively high target point coverage and slight variance, which indicates that these algorithms play a role in the coverage optimization of wireless sensors. Specifically, the optimization results of WOA and DOA are similar, ranging from 75% to 80%, while the optimization effect of WOA is slightly higher, reaching 79.68%. However, the improved WOA in this paper achieves the current best optimization results, with a coverage rate of 90.97%, higher than the second place, 11.29% of the original WOA. WOA-LFGA has the lowest variance from the second evaluation index, and its value is 0.0019. That is, the algorithm has the highest stability. The improved WOA algorithm has certain advantages over the other five algorithms in terms of performance, combining the two evaluation indexes. From the perspective of coverage efficiency, WOA-LFGA also achieves the highest node coverage efficiency among the optimized algorithms, which fully demonstrates that the algorithm has lower node redundancy and a more even distribution of nodes in the area.

As seen in Figure 5, the WOA-LFGA proposed in this paper not only reaches the highest coverage rate but also has the fastest convergence rate. When the number of iterations reaches about 30% of the maximum number of iterations, it has reached the optimum. Although BWO is an excellent algorithm, its performance could be better for the problems proposed in this paper. The optimization effect of SMA and AOA is similar, and the coverage rate calculated by them hardly changes during the iteration process. Although the final coverage rate of WOA and DOA can reach nearly 80%, their convergence rate is slower than that of WOA-LFGA, and they need to iterate more than 60% to get close to the maximum coverage rate. The improved WOA in this paper is also superior to other algorithms in terms of convergence speed and has strong practicability and effectiveness in practical applications.

Using the sensor node configuration obtained from Figure 4, the Prim algorithm [36] was employed to generate a corresponding minimum spanning tree between the nodes, which was subsequently used to depict the communication network among the monitoring nodes, as presented in Figure 6.

From the perspective of communication distance uniformity and as observed in the results, the WOA-LFGA algorithm outperforms the other five compared algorithms. Moreover, the optimized communication network generated by the WOA-LFGA algorithm features more convergence nodes located near the edge, which is conducive to shortening the distance and saving energy consumption between the nodes during data transmission. Overall, in the process of node deployment, all six algorithms are capable of optimizing the placement of the nodes, thereby enhancing the coverage of the network. However, the network coverage optimized by the WOA-LFGA algorithm demonstrates the highest level of performance, with a more uniform distribution of the nodes. This contributes to improving the reliability of the overall network and reducing energy consumption during data transmission, thus extending the working time of the network.

By varying the number of sensor nodes N deployed in the above experiment, we investigate its impact on the network coverage. Specifically, we discuss the variation of network coverage with N ranging from 10 to 30 with a step size of 5. The experimental results are presented in Figure 7 and Table 9.

The figure clearly demonstrates the trend of network coverage variation with the change in the number of sensor nodes. Specifically, when the number of nodes is 20 or less, the difference in coverage between the different algorithms is not significant. However, it gradually becomes apparent after this threshold. From the graph, it is evident that WOA-LFGA can achieve a higher network coverage than the other algorithms with the same number of nodes. Furthermore, the curve indicates that WOA-LFGA has the fastest growth rate of coverage with the increasing number of nodes, demonstrating strong competitiveness compared to the other algorithms.

#### 3.2.2. Comparison of WOA-LFGA with Different Modified WOA

In this subsection, the effectiveness of WOA-LFGA is measured by comparison with CWOA, WOABAT [37], RDWOA [38], WOAmM [39], EGE-WOA, where CWOA uses tent mapping. The parameter settings of the comparison algorithm are taken from the corresponding literature. In this experiment, the algorithm proposed in this paper is compared with the above five population intelligent algorithms, and the program operation results are shown in Table 10 and Figure 8.

It can be seen from Table 10 and Figure 9 that WOA-LFGA significantly improves the coverage optimization of WSN compared with CWOA, WOABAT, RDWOA, WOAmM, and EGE-WOA. In this experiment, the performance effect of EGE-WOA could be better. The coverage rate of EGE-WOA is below 60%, while that of CWOA is 68.34%, slightly higher than that of EGE-WOA. The coverage rate of WOA-BAT, RDWOA, and WOAmM reached an average of about 80%, or individual rates of 78.05%, 81.98%, and 81.24%, respectively. The average coverage rate of WOA-LFGA proposed in this paper is 90.97%, which achieves relatively adequate coverage and is 9% higher than the second place. WOA-LFGA has the lowest variance from the second evaluation index, and the algorithm stability is the best among several improved WOAs. Overall, WOA-LFGA has advantages over the other five algorithms’ overall performances.

Using the sensor node configuration obtained from Figure 8, the Prim algorithm was employed to generate a corresponding minimum spanning tree between the nodes, which was subsequently used to depict the communication network among the monitoring nodes, as presented in Figure 10.

From Figure 10, it is evident that WOA-LFGA still achieves more even communication distances compared with other optimized algorithms. This reduces the transmission power for information exchange, thus saving energy and extending the usage time of the entire network.

By varying the number of sensor nodes N deployed in the above experiment, we investigate its impact on the network coverage. Specifically, we discuss the variation of network coverage with N ranging from 10 to 30 with a step size of 5. The experimental results are presented in Figure 11 and Table 11.

Clearly, the following algorithms did not show much difference in optimization performance when the number of sensor nodes was less than 20. However, as the number exceeded 20, the advantage of WOA-LFGA gradually became apparent. With the same number of nodes, WOA-LFGA can better deploy wireless sensor nodes and maximize the coverage of the entire network. In terms of standard deviation, regardless of the number of sensor nodes, WOA-LFGA always has the smallest standard deviation, indicating that this algorithm has the best stability and the fastest growth rate. All these indicators together demonstrate the strong competitiveness of WOA-LFGA.

### 3.3. WOA-LFGA for WSN Coverage Practical Application

With the unprecedented development of big data, the widespread adoption of the fifth-generation mobile communication technology (5G) has accelerated. Currently, telecommunications operators worldwide are gradually rolling out 5G networks, and the development and application prospects of 5G technology are extremely promising. It can support a larger number of device connections and can contribute to the development of the Internet of Things (IoT) and the construction of smart cities. In this section, we apply the wireless sensor coverage optimization problem to real-life scenarios. Taking Jilin Jianzhu University as an example, as shown in Figure 12a, its outline can be abstracted as an irregular pentagon. For the sake of convenience in calculations, we rotate the shape counterclockwise by 90 degrees, as depicted in Figure 12b.

Equation (25) imposes constraints on the new boundary range.
(25)$$\left\{\begin{array}{ll} 0.325x<y<0.077x+950, & 0<x\leq 260\\ 0.325x<y<-0.281x+1043.125, & 261<x\leq 400\\ 4.167x-1536.667<y<-0.281x+1043.125, & 401<x\leq 580 \end{array}\right.$$

In this experiment, we tested the coverage of 13 target points within the aforementioned pentagonal area, in which the coverage radius of each target point was 100 m. To ensure that the experimental conclusions are more persuasive, we conducted 30 experiments, with 200 iterations for each experiment. The parameter settings of the experiment are shown in Table 12.

#### 3.3.1. Comparison of WOA-LFGA with Other Basic Algorithms

In this subsection, the effectiveness of WOA-LFGA is measured by comparison with SMA, DOA, AOA, BWO, and WOA. Parameter Settings of the comparison algorithm are all taken from the corresponding literature. In this experiment, the algorithm proposed in this paper is compared with the above five population intelligent algorithms, and the program operation results are shown in Table 13.

It can be seen from Table 13 that, compared with SMA, DOA, AOA, BWO, and WOA, WOA-LFGA has significantly improved the coverage optimization of the WSN. In comparison, DOA, AOA, and BWO demonstrate better optimization performance, achieving coverage rates of over 50%. Both SMA and WOA also provided feasible solutions for wireless sensor coverage in this experiment. These algorithms have proven their efficacy in practical applications. Specifically, AOA and BWO yield similar optimization results, with coverage rates around 52%. DOA yields slightly higher optimization results, surpassing 53% coverage. However, the improved WOA algorithm in this study achieved the best optimization results, with an average coverage rate of 83.77%, surpassing the second-ranked algorithm by 30.71%. This is mainly due to WOA-LFGA’s different boundary handling strategy and excellent global search capabilities. In terms of variance, WOA-LFGA has the lowest variance, with a value of 0.0035, indicating its good stability. In terms of coverage efficiency, WOA-LFGA also achieves the highest node coverage efficiency among all algorithms, demonstrating lower node redundancy and a more uniform distribution of nodes in the area. Considering these three evaluation indexes, the improved WOA algorithm has certain advantages over the other five algorithms in terms of performance.

As seen in Figure 13, the WOA-LFGA proposed in this paper not only reaches the highest coverage rate but also has the fastest convergence rate. During the iteration process, it remained in a state of growth. Specifically, the growth was rapid in the first 30% of the iterations, but then slowed. The optimization effects of DOA and AOA are similar, as they show minimal noticeable growth during the iteration process. BWO exhibits relatively rapid growth, but the results are not significantly different from DOA. SMA and WOA are also excellent algorithms, but their performance in this experiment was not very satisfactory. The improved WOA presented in this paper outperforms other algorithms in terms of convergence speed and accuracy and demonstrates strong practicality and effectiveness in practical applications.

#### 3.3.2. Comparison of WOA-LFGA with Different Modified WOA

In this subsection, the effectiveness of WOA-LFGA is measured by comparison with CWOA, WOABAT, RDWOA, WOAmM, and EGE-WOA, where CWOA uses tent mapping. The parameter settings of the comparison algorithm are taken from the corresponding literature. In this experiment, the algorithm proposed in this paper is compared with the above five population intelligent algorithms, and the program operation results are shown in Table 14.

It can be seen from Table 14 and Figure 14 that WOA-LFGA significantly improves the coverage optimization of WSN compared with CWOA, WOABAT, RDWOA, WOAmM, and EGE-WOA. In this experiment, the average coverage obtained by different improved whale algorithms varies. Among them, the evaluation coverage of WOABAT and WOAmM is less than 40%, while the average coverage obtained by EGE-WOA is slightly higher, reaching nearly 44%. The coverage of CWOA and RDWOA can exceed 50%, while the average coverage of the WOA-LFGA proposed in this article can reach 83.77%, which is 30.37% higher than the second place. The variance of WOA-LFGA is the smallest among several algorithms, which fully demonstrates that the stability of the algorithm is better than other algorithms. WOA-LFGA also has the highest coverage efficiency, which fully proves that the algorithm has a lower node redundancy and a more uniform regional node distribution. In summary, WOA-LFGA outperforms other algorithms in practical applications.

## 4. Conclusions

As IoT technology continues to advance, the internet of everything is becoming a reality. The rapid development of 5G technology will push various industries towards greater intelligence and efficiency. Smart cities, transportation, healthcare, and classrooms are increasingly integrated into people’s lives. The widespread application of IoT relies heavily on the rapid development of wireless sensors, which requires the reasonable deployment of sensor nodes within the monitored space, enabling fewer nodes to achieve greater coverage. This paper reviews solutions from international scholars and their teams that have addressed the coverage optimization problem of wireless sensors and conducts in-depth research on WSN coverage optimization using swarm intelligence algorithms.

This paper proposes the WOA-LFGA based on Lévy flight and genetic optimization mechanisms to comprehensively improve the effectiveness of the whale optimization algorithm. The algorithm has been successfully applied to 35 benchmark test functions and wireless sensor coverage problems. Comparative analysis of experimental simulation results reveals that the WOA-LFGA exhibits excellent global and local search abilities. Tests with single-mode and multi-mode reference functions demonstrate significant improvements in the algorithm’s convergence speed and accuracy, and its stability and ability to escape local optima are highly competitive compared with other intelligent optimization algorithms. Moreover, when applied to WSN coverage optimization problems and compared with other intelligent optimization algorithms, WOA-LFGA yields better optimization results, substantially improving coverage, convergence, and algorithm stability. Based on the experimental results presented in this paper, WOA-LFGA demonstrates strong competitiveness in intelligent optimization. Its application to other practical problems will become a more critical research direction in the future.

## Figures and Tables

**Figure 1 biomimetics-08-00354-f001:**
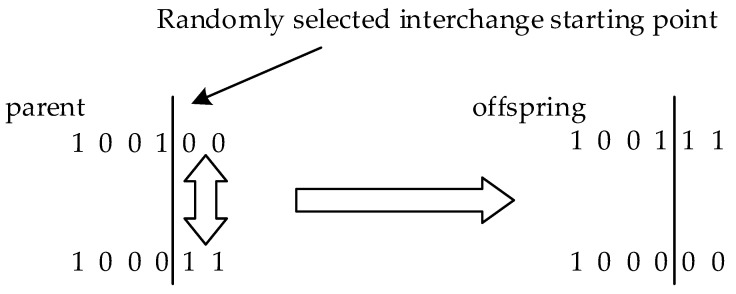
The crossover strategy in the genetic algorithm.

**Figure 2 biomimetics-08-00354-f002:**
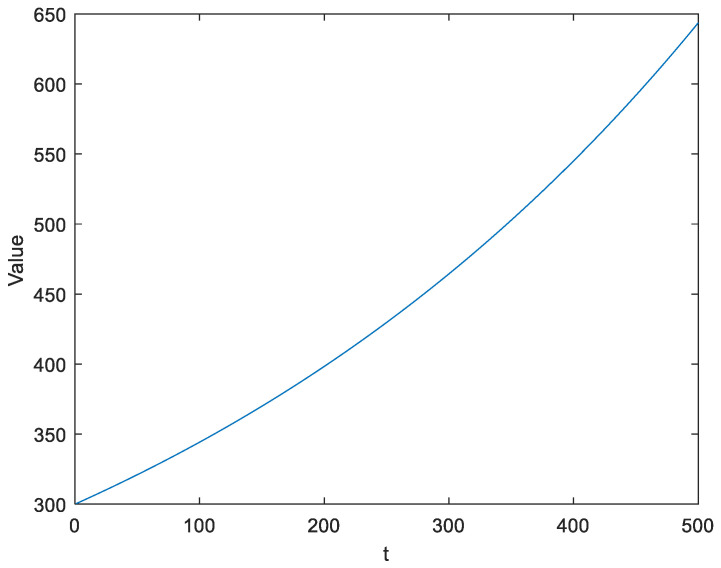
The curve characteristics of Equation (21).

**Figure 3 biomimetics-08-00354-f003:**
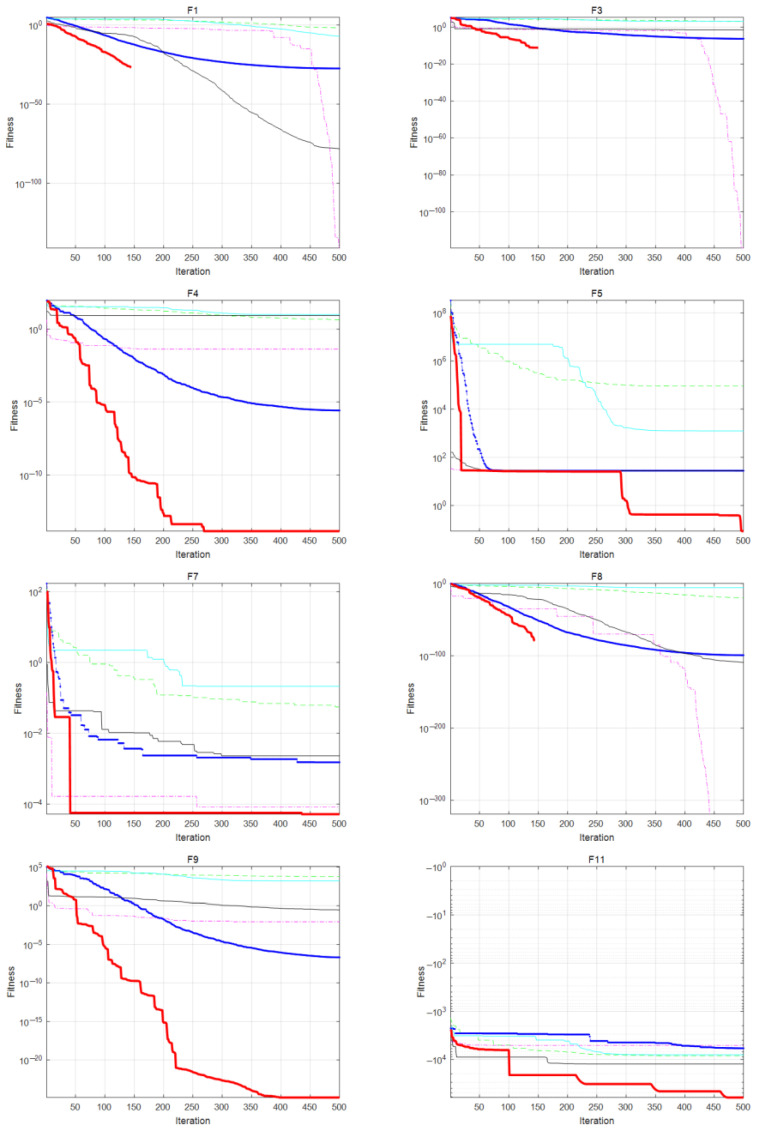
Comparison of convergence curves of WOA-LFGA and other algorithms obtained in some of the benchmark problems.

**Figure 4 biomimetics-08-00354-f004:**
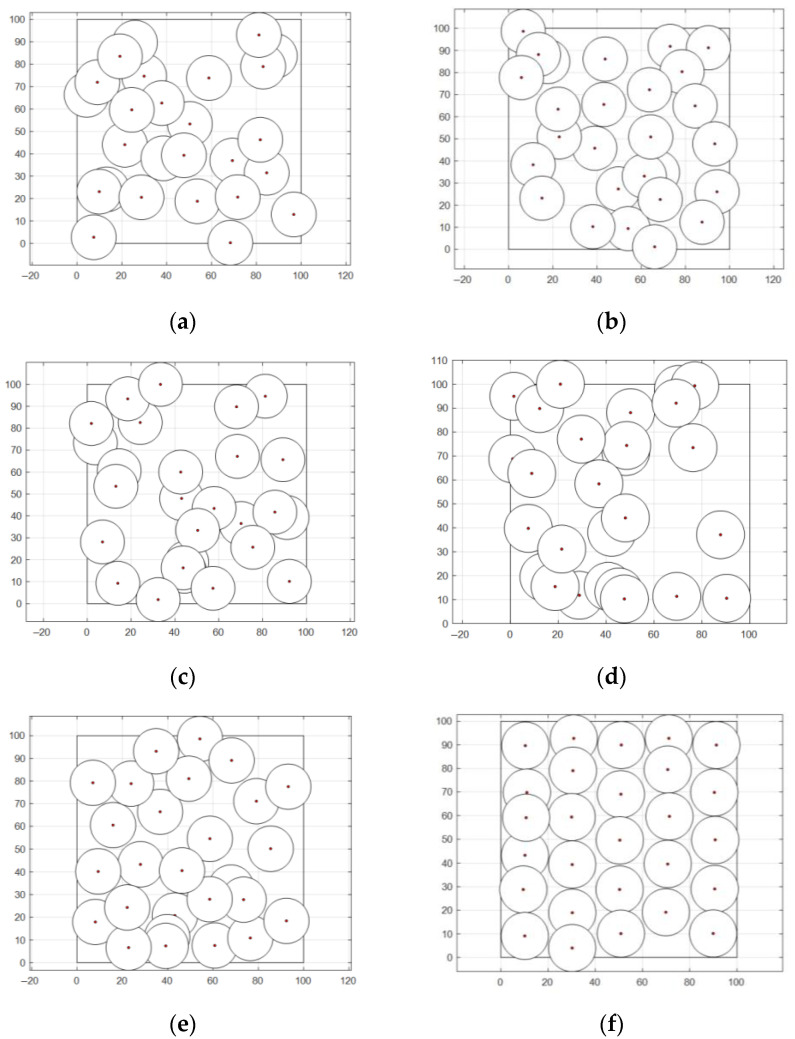
Node coverage distribution diagram. (**a**) SMA, (**b**) DOA, (**c**) AOA, (**d**) BWO, (**e**) WOA, (**f**) WOA-LFGA.

**Figure 5 biomimetics-08-00354-f005:**
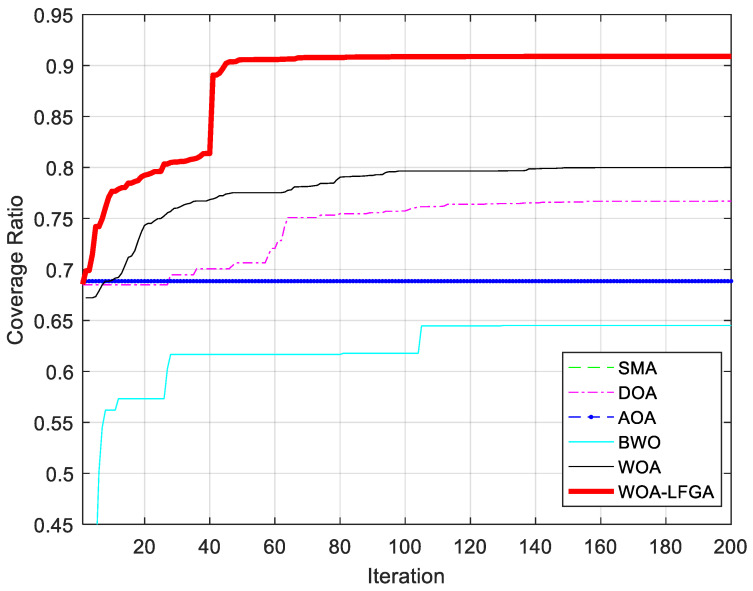
Comparison of convergence curves of WOA-LFGA and other basic algorithms obtained in WSN coverage optimization problem.

**Figure 6 biomimetics-08-00354-f006:**
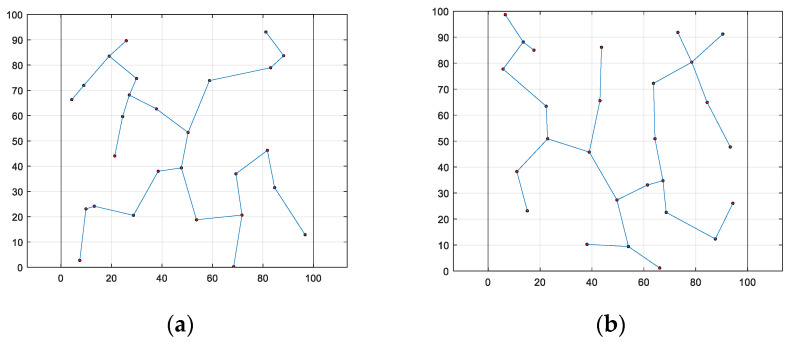
The composition of sensor nodes. (**a**) SMA, (**b**) DOA, (**c**) AOA, (**d**) BWO, (**e**) WOA, (**f**) WOA-LFGA.

**Figure 7 biomimetics-08-00354-f007:**
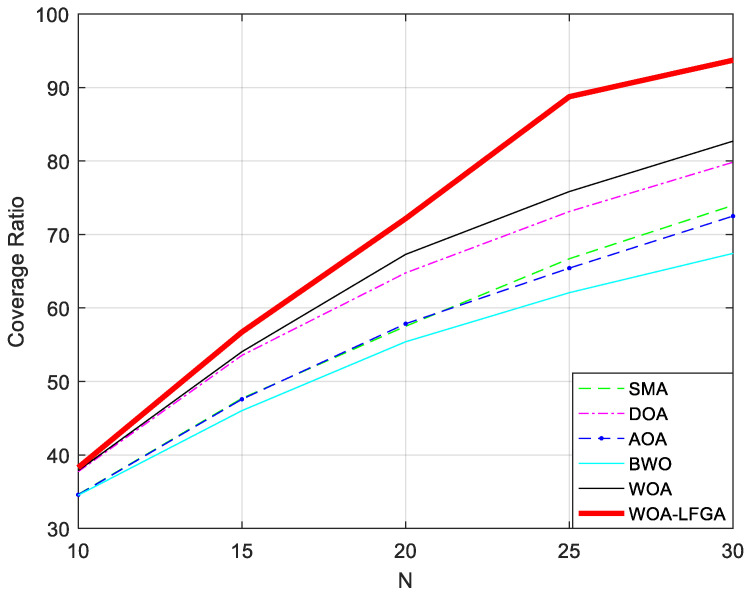
The impact of the number of sensor nodes on the network coverage between WOA-LFGA and other basic algorithms.

**Figure 8 biomimetics-08-00354-f008:**
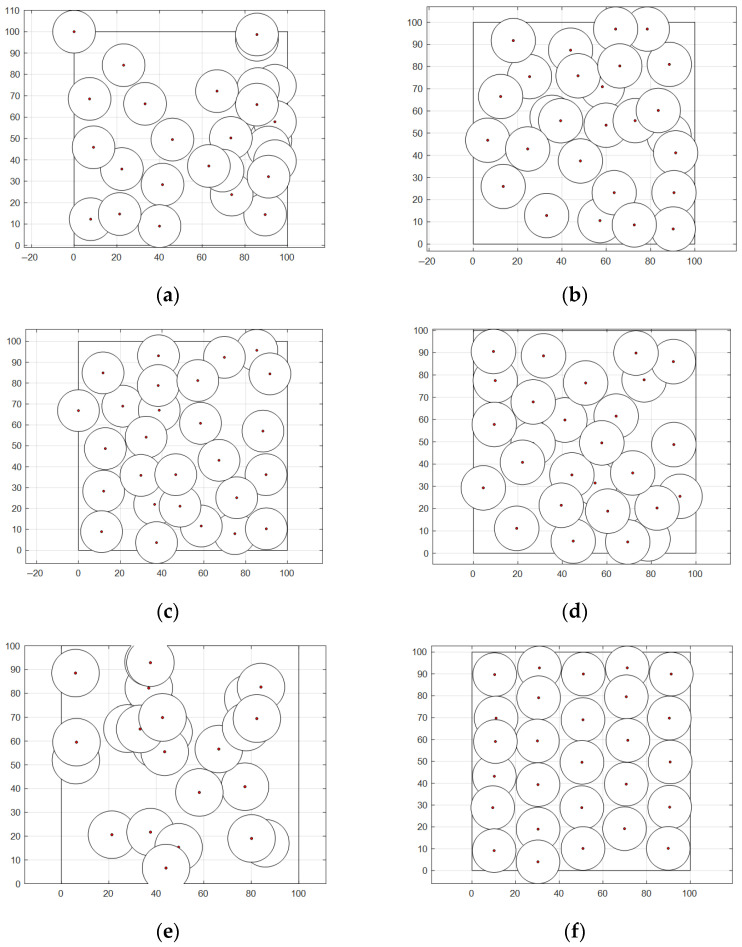
Node coverage distribution diagram. (**a**) CWOA, (**b**) WOABAT, (**c**) RDWOA, (**d**) WOAmM, (**e**) EGE-WOA, (**f**) WOA-LFGA.

**Figure 9 biomimetics-08-00354-f009:**
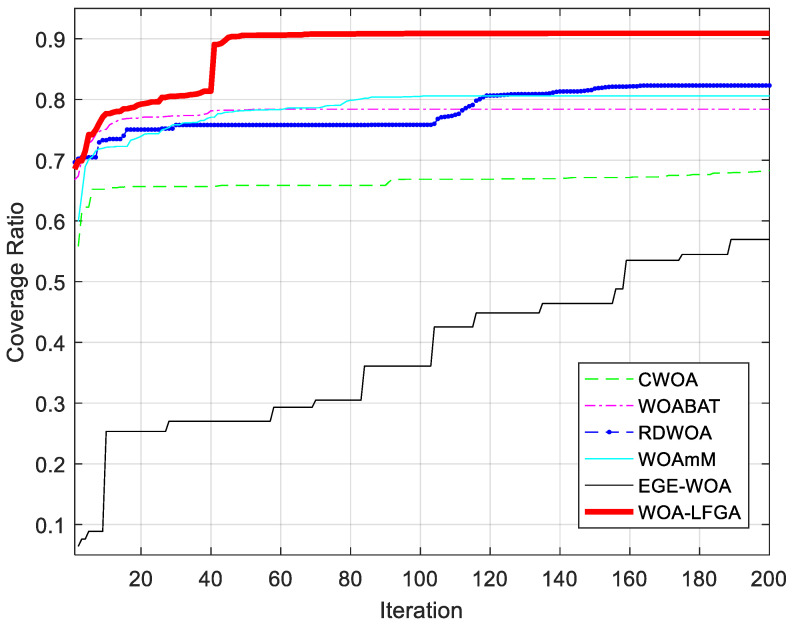
Comparison of convergence curves of WOA-LFGA and different modified WOA obtained in WSN coverage optimization problem.

**Figure 10 biomimetics-08-00354-f010:**
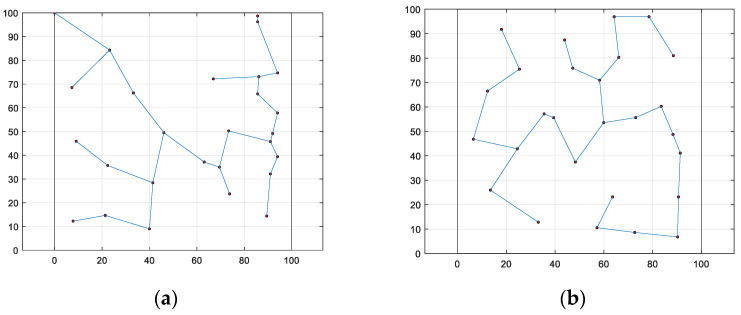
The composition of sensor nodes. (**a**) CWOA, (**b**) WOABAT, (**c**) RDWOA, (**d**) WOAmM, (**e**) EGE-WOA, (**f**) WOA-LFGA.

**Figure 11 biomimetics-08-00354-f011:**
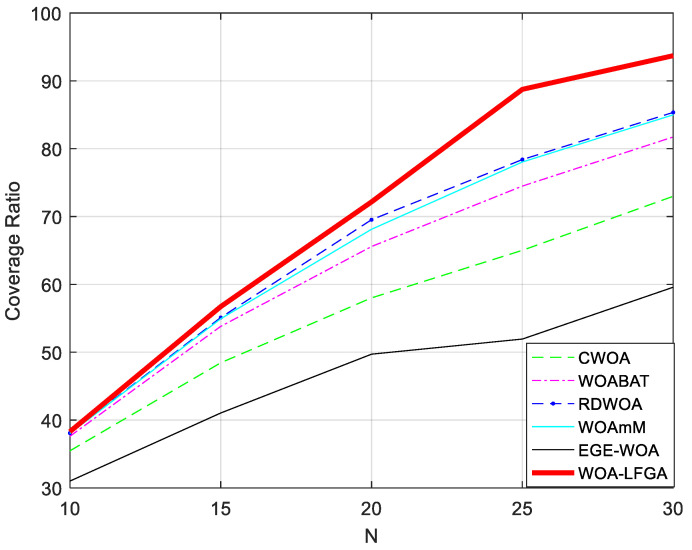
The impact of the number of sensor nodes on the network coverage between WOA-LFGA and different modified WOA.

**Figure 12 biomimetics-08-00354-f012:**
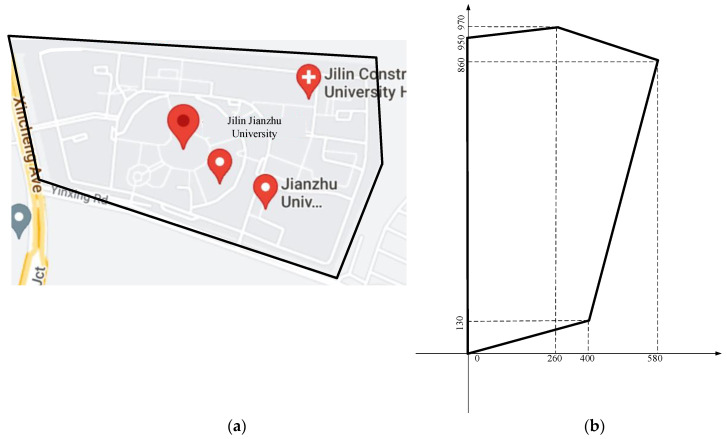
Jilin Jianzhu University. (**a**) Map of Jilin Jianzhu University, (**b**) Abstract of the Outline of Jilin Jianzhu University.

**Figure 13 biomimetics-08-00354-f013:**
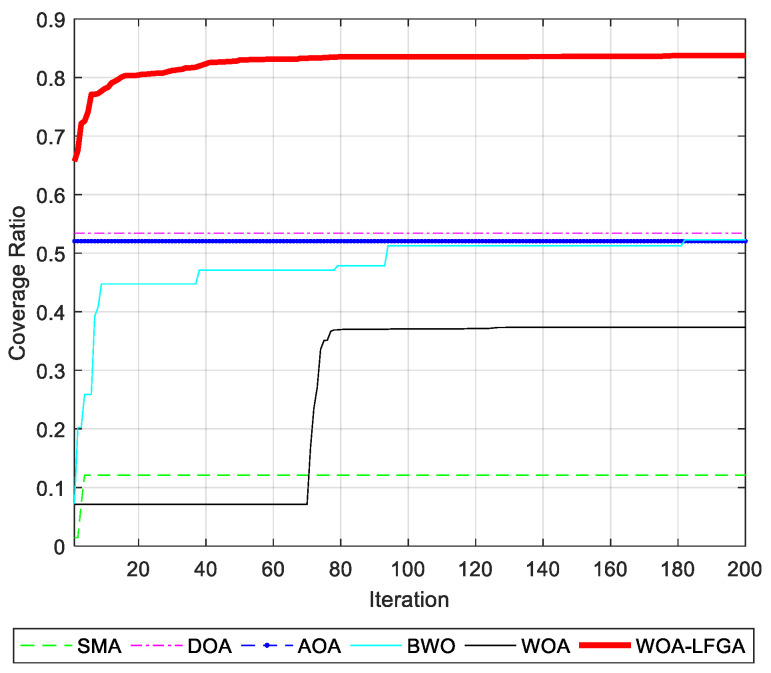
Comparison of convergence curves of WOA-LFGA and other basic algorithms obtained in WSN coverage practical application.

**Figure 14 biomimetics-08-00354-f014:**
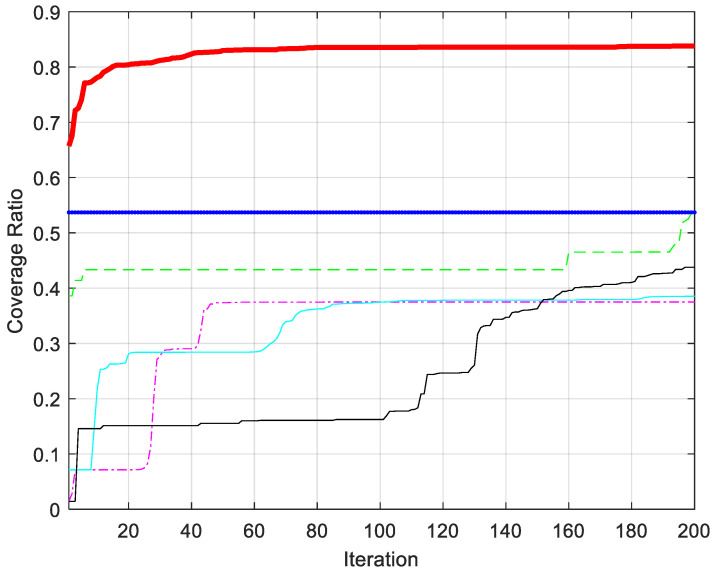
Comparison of convergence curves of WOA-LFGA and different modified WOA obtained in WSN coverage practical application.

**Table 1 biomimetics-08-00354-t001:** Description of unimodal benchmark functions.

Function	D	Range	f_min_
$F_{1}\left(x\right)=\overset{D}{\underset{i=1}{\sum }}x_{i}^{2}$	30	[−100, 100]	0
$F_{2}\left(x\right)=\overset{D}{\underset{i=1}{\sum }}\left|x_{i}\right|+\overset{D}{\underset{i=1}{\prod }}\left|x_{i}\right|$	30	[−10, 10]	0
$F_{3}\left(x\right)=\overset{D}{\underset{i=1}{\sum }}{\left(\overset{i}{\underset{j-1}{\sum }}x_{j}\right)}^{2}$	30	[−100, 100]	0
$F_{4}\left(x\right)={max}_{i}\{|x_{i}|,1\leq i\leq D\}$	30	[−100, 100]	0
$F_{5}\left(x\right)=\overset{D-1}{\underset{i=1}{\sum }}\left[100{\left(x_{i+1}-x_{i}^{2}\right)}^{2}+{\left(x_{i}-1\right)}^{2}\right]$	30	[−30, 30]	0
$F_{6}\left(x\right)=\overset{D}{\underset{i=1}{\sum }}{\left(\left[x_{i}+0.5\right]\right)}^{2}$	30	[−100, 100]	0
$F_{7}\left(x\right)=\overset{D}{\underset{i=1}{\sum }}ix_{i}^{4}+random\left[0,\;1\right)$	30	[−1.28, 1.28]	0
$F_{8}\left(x\right)=\overset{D}{\underset{i=1}{\sum }}{\left|x_{i}\right|}^{i+1}$	30	[−1, 1]	0
$F_{9}\left(x\right)=\overset{D}{\underset{i=1}{\sum }}{\left(\overset{D}{\underset{j=1}{\sum }}x_{j}\right)}^{2}$	30	[−100, 100]	0
$F_{10}\left(x\right)=\overset{D}{\underset{i=1}{\sum }}x_{i}^{2}+{\left(\overset{D}{\underset{i=1}{\sum }}0.5ix_{i}\right)}^{2}+{\left(\overset{D}{\underset{i=1}{\sum }}0.5ix_{i}\right)}^{4}$	30	[−5, 10]	0

**Table 2 biomimetics-08-00354-t002:** Description of multimodal benchmark functions.

Function	D	Range	f_min_
$F_{11}\left(x\right)=\overset{D}{\underset{i=1}{\sum }}-x_{i}sin\left(\sqrt{\left|x_{i}\right|}\right)$	30	[−500, 500]	−418.98 × D
$F_{12}\left(x\right)=1+\overset{D}{\underset{i=1}{\sum }}sin^{2}\left(x_{i}\right)-\exp\left(\overset{D}{\underset{i=1}{\sum }}x_{i}^{2}\right)$	30	[−10, 10]	0
$F_{13}\left(x\right)=0.5\overset{D}{\underset{i=1}{\sum }}\left(x_{i}^{4}-16x_{i}^{2}+5x_{i}\right)$	30	[−5, 5]	−39.166 × D
$F_{14}\left(x\right)=\overset{D}{\underset{i=1}{\sum }}[x_{i}^{2}-10\;cos\left(2\pi x_{i}\right)+10]$	30	[−5.12, 5.12]	0
$F_{15}\left(x\right)=-20exp\left(-0.2\sqrt{\frac{1}{n}\overset{D}{\underset{i=1}{\sum }}x_{i}^{2}}\right)-exp\left(\frac{1}{n}\overset{D}{\underset{i=1}{\sum }}\cos\left(2\pi x_{i}\right)\right)+20+e$	30	[−32, 32]	0
$\begin{array}{c} F_{16}\left(x\right)=\frac{1}{4000}\overset{D}{\underset{i=1}{\sum }}x_{i}^{2}-\overset{D}{\underset{i=1}{\prod }}cos\left(\frac{x_{i}}{\sqrt{i}}\right)+1 \end{array}$	30	[−600, 600]	0
$\begin{array}{c} F_{17}\left(x\right)=\left(\overset{D}{\underset{i=1}{\sum }}sin^{2}\left(x_{i}\right)-exp\left(-\overset{D}{\underset{i=1}{\sum }}x_{i}^{2}\right)\right)\\ exp\left(-\overset{D}{\underset{i=1}{\sum }}sin^{2}\sqrt{\left|x_{i}\right|}\right) \end{array}$	30	[−10, 10]	−1
$\begin{array}{l} F_{18}\left(x\right)=\frac{\pi }{D}\{10\sin\left(\pi y_{1}\right)+\overset{D-1}{\underset{i=1}{\sum }}{\left(y_{i}-1\right)}^{2}\left[1+10\sin^{2}\left(\pi y_{i+1}\right)\right]\\ \;\;\;\;\;\;\;\;\;\;\;\;\;\;\;\;\;+{\left(y_{n}-1\right)}^{2}\}+\overset{D}{\underset{i=1}{\sum }}u\left(x_{i},\;10,\;100,\;4\right) \end{array}$	30	[−50, 50]	0
$F_{19}\left(x\right)=0.1\{\sin^{2}\left(3\pi x_{1}\right)+\overset{D}{\underset{i=1}{\sum }}{\left(x_{i}-1\right)}^{2}\left[1+\sin^{2}\left(3\pi x_{i}+1\right)\right]$ $+{\left(x_{n}-1\right)}^{2}\left[1+sin^{2}\left(2\pi x_{D}\right)\right]\}+\overset{D}{\underset{i=1}{\sum }}u\left(x_{i},\;5,\;100,\;4\right)$	30	[−50, 50]	0
$F_{20}\left(x\right)={\left(\frac{1}{500}+\overset{25}{\underset{j=1}{\sum }}\frac{1}{j+\sum_{i=1}^{2}{\left(x_{i}-a_{ij}\right)}^{6}}\right)}^{-1}$	2	[−65, 65]	1
$F_{21}\left(x\right)=\overset{11}{\underset{i=1}{\sum }}{\left[a_{i}-\frac{x_{1}\left(b_{i}^{2}+b_{i}x_{2}\right)}{b_{i}^{2}+b_{i}x_{3}+x_{4}}\right]}^{2}$	4	[−5, 5]	0.00030
$F_{22}\left(x\right)=4x_{1}^{2}-2.1x_{1}^{4}+\frac{1}{3}x_{1}^{6}+x_{1}x_{2}-4x_{2}^{2}+4x_{2}^{4}$	2	[−5, 5]	−10.316
$F_{23}\left(x\right)={\left(x_{2}-\frac{5.1}{4\pi^{2}}x_{1}^{2}+\frac{5}{\pi }x_{1}-6\right)}^{2}+10\left(1-\frac{1}{8\pi }\right)\cos x_{1}+10$	2	[−5, 5]	0.398
$F_{24}\left(x\right)=[1+{\left(x_{1}+x_{2}+1\right)}^{2}(19-14x_{1}+3x_{1}^{2}-14x_{2}$ $+6x_{1}x_{2}+3x_{2}^{2})]\;\times [30+{\left(2x_{1}-3x_{2}\right)}^{2}$ $\times \left(18-32x_{1}+12x_{1}^{2}+48x_{2}-36x_{1}x_{2}+27x_{2}^{2}\right)]$	2	[−2, 2]	−3
$F_{25}\left(x\right)=-\overset{4}{\underset{i=1}{\sum }}c_{i}exp\left(-\overset{3}{\underset{j=1}{\sum }}a_{ij}{\left(x_{j}-p_{ij}\right)}^{2}\right)$	3	[1, 3]	−3.86
$F_{26}\left(x\right)=-\overset{4}{\underset{i=1}{\sum }}c_{i}exp\left(-\overset{6}{\underset{j=1}{\sum }}a_{ij}{\left(x_{j}-p_{ij}\right)}^{2}\right)$	6	[0, 1]	−3.32
$F_{27}\left(x\right)=-\overset{5}{\underset{i=1}{\sum }}{\left[\left(X-a_{i}\right){\left(X-a_{i}\right)}^{T}+c_{i}\right]}^{-1}$	4	[0, 10]	−10.1532
$F_{28}\left(x\right)=-\overset{7}{\underset{i=1}{\sum }}{\left[\left(X-a_{i}\right){\left(X-a_{i}\right)}^{T}+c_{i}\right]}^{-1}$	4	[0, 10]	−10.4028
$F_{29}\left(x\right)=-\overset{10}{\underset{i=1}{\sum }}{\left[\left(X-a_{i}\right){\left(X-a_{i}\right)}^{T}+c_{i}\right]}^{-1}$	4	[0, 10]	−10.5363

**Table 3 biomimetics-08-00354-t003:** Description of composite benchmark functions.

Function	D	Range	f_min_
F_30_(CF1): f_1_, f_2_, f_3_,…, f_10_ = Sphere Function [σ_1_, σ_2_, σ_3_,…, σ_10_] = [1, 1, 1,…, 1] [λ_1_, λ_2_, λ_3_,…, λ_10_] = [5/100, 5/100, 5/100,…, 5/100]	10	[−5, 5]	0
F_31_(CF2): f_1_, f_2_, f_3_,…, f_10_ = Griewank’s Function [σ_1_, σ_2_, σ_3_,…, σ_10_] = [1, 1, 1,…, 1] [λ_1_, λ_2_, λ_3_,…, λ_10_] = [5/100, 5/100, 5/100,…, 5/100]	10	[−5, 5]	0
F_32_(CF3): f_1_, f_2_, f_3_,…, f_10_ = Griewank’s Function [σ_1_, σ_2_, σ_3_,…, σ_10_] = [1, 1, 1,…, 1] [λ_1_, λ_2_, λ_3_,…, λ_10_] = [1, 1, 1,…, 1]	10	[−5, 5]	0
F_33_(CF4): f_1_, f_2_ = Ackley’s Function, f_3_, f_4_ = Rastrigin’s Function, f_5_, f_6_ = Weierstrass Function, f_7_, f_8_ = Griewank’s Function, f_9_, f_10_ = Sphere’s Function [σ_1_, σ_2_, σ_3_,…, σ_10_] = [1, 1, 1,…, 1] [λ_1_, λ_2_, λ_3_,…, λ_10_] = [5/32, 5/32, 1, 1, 5/0.5, 5/0.5, 5/100, 5/100, 5/100, 5/100]	10	[−5, 5]	0
F_34_(CF5): f_1_, f_2_ = Rastrigin’s Function, f_3_, f_4_ = Weierstrass Function, f_5_, f_6_ = Griewank’s Function, f_7_, f_8_ = Ackley’s Function, f_9_, f_10_ = Sphere’s Function [σ_1_, σ_2_, σ_3_,…, σ_10_] = [1, 1, 1,…, 1] [λ_1_, λ_2_, λ_3_,…, λ_10_] = [1/5, 1/5, 5/0.5, 5/0.5, 5/100, 5/100, 5/32, 5/32, 5/100, 5/100]	10	[−5, 5]	0
F_35_(CF6): f_1_, f_2_ = Rastrigin’s Function, f_3_, f_4_ = Weierstrass Function, f_5_, f_6_ = Griewank’s Function, f_7_, f_8_ = Ackley’s Function, f_9_, f_10_ = Sphere’s Function [σ_1_, σ_2_, σ_3_,…, σ_10_] = [0.1, 0.2, 0.3, 0.4, 0.5, 0.6, 0.7, 0.8, 0.9, 1] [λ_1_, λ_2_, λ_3_,…, λ_10_] = [0.1 × 1/5, 0.2 × 1/5, 0.3 × 5/0.5, 0.4 × 5/0.5, 0.5 × 5/100, 0.6 × 5/100, 0.7 × 5/32, 0.8 × 5/32, 0.9 × 5/100, 1 × 5/100]	10	[−5, 5]	0

**Table 4 biomimetics-08-00354-t004:** Comparison of optimization results obtained for the unimodal and multimodal benchmark functions.

	PSO		AOA		GWO		SSA		WOA		WOA-LFGA	
	ave	std	ave	std	ave	std	ave	std	ave	std	ave	std
F1	0.01145	0.016214	1.82 × 10^−20^	9.99 × 10^−20^	2.71 × 10^−27^	7.04 × 10^−27^	1.42 × 10^−07^	1.62 × 10^−07^	2.31 × 10^−71^	1.14 × 10^−70^	0	0
F2	2.020543	4.065539	0	0	1.08 × 10^−16^	8.93 × 10^−17^	2.285371	1.666859	1.07 × 10^−50^	4.58 × 10^−50^	0	0
F3	2444.118	1926.835	0.003659	0.007401	9.85 × 10^−06^	1.90 × 10^−05^	1382.524	777.8525	71.50822	172.5122	0	0
F4	7.036386	1.311499	0.025943	0.019751	8.16 × 10^−07^	8.64 × 10^−07^	11.60046	3.603777	1.293207	1.217586	1.65 × 10^−10^	7.94 × 10^−10^
F5	237.9364	552.168	28.43077	0.241825	27.0996	0.744425	358.5001	543.5524	27.72318	0.381725	20.77285	10.26487
F6	0.008698	0.014406	3.18966	0.252549	0.767048	0.393704	2.80 × 10^−07^	5.94 E−07	0.263031	0.199383	0.070958	0.122792
F7	0.049637	0.017275	6.93 × 10^−05^	6.73 E−05	0.001663	0.0008	0.190837	0.075292	0.003031	0.002759	0.001748	0.003825
F8	1.62 × 10^−18^	7.45 × 10^−18^	0	0	1.60 × 10^−94^	8.74 × 10^−94^	1.60 × 10^−06^	1.04 × 10^−06^	8.07 × 10^−101^	4.42 × 10^−100^	0	0
F9	3715.167	3804.074	0.006306	0.015188	1.00 × 10^−05^	1.44 × 10^−05^	1543.173	827.1876	139.0415	349.0566	3.84 × 10^−26^	1.99 × 10^−25^
F10	135.1749	86.41104	278.754	50.12035	3.35 × 10^−07^	7.80 × 10^−07^	43.32593	15.77741	25.84223	104.563	6.12 × 10^−17^	3.35 × 10^−16^
F11	−8588.58	743.6667	−5347.08	428.9775	−5856.44	736.0021	−7429.88	767.0725	−10327.8	1815.032	−62304.4	2.22 × 10^−11^
F12	1.85834	0.705254	0	0	2.08691	2.001494	1	1.48 × 10^−09^	0.129003	0.407659	0	0
F13	−1010.53	32.04003	−488.895	65.78818	−906.163	66.86702	−999.69	41.44037	−1173.67	3.427174	−1174.98	0.005266
F14	54.17668	12.63687	0	0	1.948371	3.150168	47.85746	15.99706	1.89 × 10^−15^	1.04 × 10^−14^	0	0
F15	0.768649	0.668676	8.88 × 10^−16^	0	1.03 × 10^−13^	1.69 × 10^−14^	2.481978	0.913383	4.20 × 10^−15^	2.46 × 10^−15^	8.88 × 10^−16^	0
F16	0.035694	0.042562	0.182622	0.131219	0.004629	0.008419	0.015976	0.00876	0.01046	0.039824	0	0
F17	7.94 × 10^−15^	4.29 × 10^−14^	7.38 × 10^−08^	6.46 × 10^−08^	1.19 × 10^−15^	3.31 × 10^−16^	2.39 × 10^−16^	1.31 × 10^−15^	−1	4.61 × 10^−17^	−1	0
F18	0.170733	0.276331	0.521644	0.051792	0.054542	0.02857	6.834328	2.62791	0.01398	0.016893	0.006835	0.019706
F19	0.156988	0.196888	2.840098	0.098464	0.628701	0.19635	13.60701	14.96327	0.278207	0.185077	0.208052	0.199385
F20	0.998004	5.83 × 10^−17^	8.2796	4.850009	3.676116	3.874222	1.295293	0.827786	2.865604	2.997616	1.687328	1.873362
F21	0.002626	0.006021	0.012879	0.022459	0.004451	0.008095	0.001558	0.003563	0.000612	0.000297	0.000362	0.000218
F22	−1.03163	6.45 × 10^−16^	−1.03163	1.30 × 10^−07^	−1.03163	2.57 × 10^−08^	−1.03163	2.67 × 10^−14^	−1.03163	9.35 × 10^−10^	−1.03163	4.91 × 10^−16^
F23	0.397887	0	0.40893	0.008738	0.397889	2.69 × 10^−06^	0.397887	3.68 × 10^−14^	0.397891	8.11 × 10^−06^	0.397887	6.37 × 10^−15^
F24	3	1.24 × 10^−15^	6.60127	9.334635	3.00005	6.62 × 10^−05^	3	2.13 × 10^−13^	3.900112	4.929503	3	1.63 × 10^−06^
F25	−3.86278	2.65 × 10^−15^	−3.85196	0.004077	−3.8615	0.00228	−3.86278	1.89 × 10^−11^	−3.85717	0.009316	−3.86278	2.14 × 10^−15^
F26	−3.26514	0.07867	−3.06929	0.075255	−3.25627	0.085463	−3.21838	0.05356	−3.25865	0.122607	−3.28633	0.055417
F27	−6.01714	3.525257	−3.58449	1.100864	−8.51535	2.579639	−8.65039	2.831563	−6.57441	2.361348	−10.1532	3.51 × 10^−14^
F28	−8.44332	3.329769	−4.01415	1.838357	−10.4014	0.000913	−8.44212	3.093598	−7.38245	2.669453	−10.4029	2.28 × 10^−13^
F29	−7.2628	3.864573	−3.45313	1.352861	−9.81322	2.238027	−8.03092	3.636316	−7.68458	2.925501	−10.5364	1.29 × 10^−12^

**Table 5 biomimetics-08-00354-t005:** Comparison of optimization results obtained for the composite benchmark functions.

	PSO		AOA		GWO		SSA		WOA		WOA-LFGA	
	ave	std	ave	std	ave	std	ave	std	ave	std	ave	std
F30	188.4598	104.2843	429.9201	122.6024	165.2451	120.3013	143.3333	138.1736	147.113	109.1869	81.9337	109.8375
F31	210.1492	147.6265	603.8082	141.238	217.9645	110.3465	193.744	119.9475	212.4452	102.3541	167.121	119.973
F32	254.4012	118.5757	739.0197	169.9494	218.669	100.6576	329.7179	239.0358	494.4398	203.5997	438.3484	132.1945
F33	497.786	191.054	853.3283	70.53408	709.6582	188.0356	630.5518	272.5582	633.3295	174.6679	576.9929	128.6628
F34	249.408	231.7561	493.5288	182.9644	187.0822	137.7849	182.7982	202.8263	206.8386	159.906	165.38	111.3664
F35	826.5022	155.8774	877.2691	66.94098	837.5018	152.0384	762.027	185.1875	824.9949	159.4651	814.3615	167.8058

**Table 6 biomimetics-08-00354-t006:** Results of test functions (F1–F19) with 30, 50, 100 and 500 dimensions.

		PSO		AOA		GWO		SSA		WOA		WOA-LFGA	
	D	ave	std	ave	std	ave	std	ave	std	ave	std	ave	std
F1	50	10.86347	14.44537	0.000863	0.001639	6.16 × 10^−20^	4.36 × 10^−20^	0.85548	1.003725	1.66 × 10^−73^	7.15 × 10^−73^	0	0
	100	2316.109	3668.931	0.021699	0.008517	1.75 × 10^−12^	1.2 × 10^−12^	1471.517	385.5454	3.39 × 10^−72^	1.51 × 10^−71^	0	0
	500	235236.4	25977.07	0.6333	0.037321	0.001453	0.000521	96418.89	5452.527	1.71 × 10^−73^	5.04 × 10^−73^	0	0
F2	50	10.53083	11.77191	2.3 × 10^−147^	1 × 10^−146^	2.51 × 10^−12^	1.32 × 10^−12^	8.895375	2.801584	2.35 × 10^−49^	1.05 × 10^−48^	0	0
	100	65.36263	22.83761	2.42 × 10^−53^	1.08 × 10^−52^	4.11 × 10^−08^	1.52 × 10^−08^	48.25606	7.948359	8.59 × 10^−50^	1.89 × 10^−49^	0	0
	500	1390.407	110.3861	0.001232	0.001668	0.010938	0.00145	541.6126	19.43979	3.87 × 10^−49^	1.68 × 10^−48^	0	0
F3	50	16884.77	4607.752	0.103386	0.097921	0.333669	0.597959	9735.95	5803.166	565.1298	762.9405	3.86 × 10^−19^	1.73 × 10^−18^
	100	101956.4	11759.16	1.127456	1.75346	636.1386	928.2477	64451	32153.11	4706.516	7708.618	3.07 × 10^−17^	1.37 × 10^−16^
	500	2764988	318150.2	33.67954	16.67636	334085.1	95550.54	1275053	728370.7	88474.96	146520.3	2.52 × 10^−12^	1.13 × 10^−11^
F4	50	17.96243	1.707676	0.046721	0.015961	0.000272	0.000202	20.63042	4.258961	2.152591	2.361506	2.43 × 10^−10^	1.07 × 10^−09^
	100	40.44695	3.397269	0.092903	0.010875	0.587254	0.433484	27.99427	2.744113	3.388053	2.958831	1.38 × 10^−10^	4.27 × 10^−10^
	500	76.60742	3.587335	0.180715	0.013151	65.33815	5.519397	40.29455	2.292022	3.380097	2.410942	9.41 × 10^−09^	3.08 × 10^−08^
F5	50	5662.017	19968.47	48.77104	0.157029	47.43632	0.947389	3276.49	5682.868	48.04747	0.403162	34.7513	20.54994
	100	203892.7	68214.38	98.87163	0.115737	97.96276	0.542074	156566.4	75343.73	98.13826	0.19119	49.03683	48.75961
	500	4.59 × 10^+08^	1.37 × 10^+08^	499.0966	0.064668	498.083	0.237754	37597520	3829547	495.8758	0.415621	161.4206	225.6898
F6	50	8.762354	7.964543	7.148222	0.382553	2.763138	0.603988	0.594813	0.590689	0.838658	0.362111	0.528771	0.43255
	100	2473.402	3667.765	18.2289	0.63456	10.5705	1.229664	1426.96	511.486	2.277557	0.810151	2.215489	1.824284
	500	229660.1	31072.21	116.0074	1.081187	92.01562	1.958327	93586.05	6284.008	19.57877	7.912018	19.30641	21.22633
F7	50	0.596402	1.806843	7.14 × 10^−05^	5.37 × 10^−05^	0.003166	0.001527	0.564758	0.128404	0.003614	0.004185	0.002136	0.003517
	100	6.536545	8.826638	6.06 × 10^−05^	5.39 × 10^−05^	0.006948	0.004253	2.843964	0.659053	0.003686	0.003096	0.001628	0.004275
	500	3707.805	687.6992	8.02 × 10^−05^	8.17 × 10^−05^	0.049075	0.012875	276.4369	53.53303	0.003276	0.004661	0.000924	0.000913
F8	50	1.44 × 10^−14^	3.85 × 10^−14^	0	0	1.86 × 10^−88^	6.35 × 10^−88^	2.19 × 10^−06^	1.69 × 10^−06^	1.2 × 10^−107^	5.5 × 10^−107^	0	0
	100	2.64 × 10^−11^	8.42 × 10^−11^	0	0	2.15 × 10^−35^	9.61 × 10^−35^	2.29 × 10^−06^	1.67 × 10^−06^	9.8 × 10^−104^	2.8 × 10^−103^	0	0
	500	1.88 × 10^−05^	4.2 × 10^−05^	0	0	0.000271	0.001131	5.71 × 10^−06^	7.21 × 10^−06^	1.3 × 10^−110^	5.7 × 10^−110^	0	0
F9	50	18886.55	5430.519	0.05631	0.049453	0.367136	0.657843	10379.31	5073.609	711.4174	1668.814	1.98 × 10^−14^	8.84 × 10^−14^
	100	106450	15037.64	1.059031	0.969092	641.6541	619.2241	43718.87	25770.16	5269.864	7375.078	1.21 × 10^−20^	5.02 × 10^−20^
	500	2696744	385835	38.4927	28.86932	328280.4	66473.19	1217109	526500.8	1625552	6833751	3.41 × 10^−18^	1.36 × 10^−17^
F10	50	624.1645	207.4525	798.6709	98.33566	0.073548	0.07947	391.2071	89.38285	46.64148	192.6738	0.004231	0.018922
	100	2536.807	451.22	2051.827	173.7202	122.6674	52.30159	1956.498	227.5388	227.8372	676.5795	4.18 × 10^−05^	0.000187
	500	24484.75	1175.066	8.84 ×10^+14^	3.54 ×10^+15^	3854.575	356.6187	10441.52	642.2583	559.5588	1632.451	500.7807	1714.189
F11	50	−12786.3	798.5763	−6730.6	555.5429	−9006.98	796.4451	−11829.8	1409.544	−17237.6	3259.934	−103841	2.99 × 10^−11^
	100	−21997.3	1611.842	−9932.03	556.0954	−16523.7	1163.17	−22109.6	1951.76	−33053	6993.32	−207681	5.97 × 10^−11^
	500	−65272.3	2572.758	−22147.4	1418.863	−53823.4	13825.98	−60450.6	5024.125	−183344	28730.99	1038407	1.19 × 10^−10^
F12	50	3.512278	1.823459	0	0	1.997852	0.767904	1.00591	0.009882	0	0	0	0
	100	8.708894	3.634477	0	0	2.969532	0.649941	3.56948	0.820859	0.052619	0.235318	0	0
	500	116.9072	9.152489	6.35 × 10^−06^	1.81 × 10^−06^	28.20371	59.83419	107.274	4.117031	5.55 × 10 ^−18^	2.48 × 10 ^−17^	0	0
F13	50	−1681.47	45.5754	−675.226	76.99906	−1352.15	90.40042	−1648.11	38.54963	−1956.64	1.398103	−1958.07	0.215703
	100	−3303.19	63.15214	−1084.03	124.9054	−2299.99	157.4946	−3023.55	71.17438	−3910.89	5.958551	−3915.88	0.840303
	500	−12380.6	261.834	−3680.81	261.3991	−7753.78	531.8809	−10816.8	224.6012	−19540.2	34.90968	−19567.5	37.75219
F14	50	119.9365	28.58934	0	0	4.178933	4.74967	88.4886	30.73374	0	0	0	0
	100	382.7355	54.84386	0	0	10.74289	7.341498	230.9327	35.07983	0	0	0	0
	500	4449.093	186.3669	5.97 × 10^−06^	5.37 × 10^−06^	70.76179	18.02281	3151.214	160.9733	4.55 × 10^−14^	2.03 × 10^−13^	0	0
F15	50	2.715587	0.484115	8.88 × 10^−16^	0	4.53 × 10^−11^	3.17 × 10^−11^	4.635025	1.206284	4.26 × 10^−15^	2.44 × 10^−15^	8.88 × 10^−16^	0
	100	6.525648	2.065837	0.000484	0.000793	1.22 × 10^−07^	4.02 × 10^−08^	10.2093	1.047667	4.26 × 10^−15^	2.7 × 10^−15^	8.88 × 10^−16^	0
	500	18.05284	0.448287	0.007914	0.000662	0.001876	0.000293	14.24981	0.224026	3.55 × 10^−15^	2.27 × 10^−15^	8.88 × 10^−16^	0
F16	50	1.059217	0.145573	1.062206	0.144497	0.003473	0.007606	0.508193	0.177961	0.008673	0.038785	0	0
	100	35.28278	50.71112	585.2056	187.6203	0.003466	0.008471	12.83264	2.844918	5.55 × 10 ^−18^	2.48 × 10 ^−17^	0	0
	500	2133.145	209.9714	10516.47	2772.351	0.004728	0.020304	867.917	65.88722	0	0	0	0
F17	50	1.53 × 10^−21^	1.16 × 10^−21^	2.82 × 10^−12^	3.22 × 10^−12^	2.6 × 10^−22^	6.62 × 10^−22^	1.47 × 10^−21^	8.59 × 10^−22^	−1	6.24 × 10^−17^	−1	0
	100	6.52 × 10^−41^	1.2 × 10^−40^	2.17 × 10^−23^	2.7 × 10^−23^	8.56 × 10^−41^	1.9 × 10^−40^	3.66 × 10^−41^	4.07 × 10^−41^	−0.85	0.366348	−1	0
	500	4.8 × 10^−177^	0	1.3 × 10^−111^	4.4 × 10^−111^	1.1 × 10^−173^	0	1.4 × 10^−182^	0	−0.7	0.470162	−1	0
F18	50	3.386843	1.228704	0.734116	0.044766	0.106871	0.047385	11.49135	2.713314	0.012943	0.007324	0.012762	0.020302
	100	2936.118	6291.82	0.901293	0.025436	0.276781	0.060778	31.0403	10.6664	0.020269	0.011114	0.016797	0.022049
	500	4.79 × 10^+08^	2.04 × 10^+08^	1.082153	0.010931	0.766924	0.058279	1530375	926662	0.024601	0.011647	0.044441	0.070781
F19	50	42.53796	20.41387	4.875282	0.080773	2.085256	0.373575	76.36419	12.04896	0.413098	0.227486	0.314442	0.409962
	100	73149.61	62122.54	9.968205	0.057886	6.84329	0.459763	9531.26	15735.96	1.139066	0.584451	1.134865	1.559223
	500	1.5 × 10^+09^	2.72 × 10^+08^	50.221	0.039006	50.05496	1.3932	34036361	9213893	7.216821	3.138501	4.851414	6.68935

**Table 7 biomimetics-08-00354-t007:** Parameters of WSN coverage optimization problem in Section 3.2.

Parameter	Value
Region size	100 m × 100 m
Sensing range	11 m
Sensor nodes number N	27
Individual number	50
Iterations	200
Test times	30

**Table 8 biomimetics-08-00354-t008:** Coverage ratio comparison of WOA-LFGA with other basic algorithms.

Method	ave	std	C
SMA	68.9237%	0.0173	0.6715
DOA	76.2457%	0.0183	0.7429
AOA	68.3437%	0.0137	0.6659
BWO	64.1613%	0.0205	0.6251
WOA	79.6813%	0.0231	0.7763
WOA-LFGA	90.9703%	0.0019	0.8863

**Table 9 biomimetics-08-00354-t009:** The variation of network coverage with different numbers of nodes between WOA-LFGA and other basic algorithms.

	N = 10		N = 15		N = 20		N = 25		N = 30	
Method	ave	std	ave	std	ave	std	ave	std	ave	std
SMA	34.63%	0.00877	47.70%	0.01028	57.51%	0.01102	66.70%	0.01215	73.96%	0.02138
DOA	37.73%	0.00657	53.55%	0.01176	64.77%	0.01743	73.12%	0.02305	79.83%	0.01749
AOA	34.59%	0.00685	47.58%	0.01183	57.85%	0.0186	65.42%	0.01425	72.50%	0.01506
BWO	34.52%	0.00919	46.03%	0.01634	55.40%	0.02252	62.08%	0.02008	67.43%	0.02664
WOA	37.87%	0.00264	54.04%	0.01373	67.29%	0.01785	75.84%	0.02531	82.70%	0.02263
WOA-LFGA	38.29%	0.00042	56.72%	0.00381	72.16%	0.00747	88.75%	0.00105	93.71%	0.00272

**Table 10 biomimetics-08-00354-t010:** Coverage ratio comparison of WOA-LFGA with different modified WOA.

Method	ave	std	C
CWOA	68.3363%	0.0263	0.6658
WOABAT	78.0493%	0.0217	0.7604
RDWOA	81.9797%	0.0171	0.7987
WOAmM	81.2440%	0.0250	0.7916
EGE-WOA	56.2650%	0.0489	0.5482
WOA-LFGA	90.9703%	0.0019	0.8863

**Table 11 biomimetics-08-00354-t011:** The variation of network coverage with different numbers of nodes between WOA-LFGA and different modified WOA.

	N = 10		N = 15		N = 20		N = 25		N = 30	
Method	ave	std	ave	std	ave	std	Ave	std	ave	std
CWOA	35.46%	0.0116	48.44%	0.0223	58.01%	0.0253	65.02%	0.0186	72.98%	0.0251
WOABAT	37.61%	0.0040	53.81%	0.0103	65.58%	0.0160	74.49%	0.0234	81.73%	0.0210
RDWOA	38.08%	0.0021	55.11%	0.0076	69.53%	0.0078	78.40%	0.0157	85.35%	0.0236
WOAmM	38.04%	0.0020	54.93%	0.0108	68.12%	0.0120	78.04%	0.0200	84.99%	0.0214
EGE-WOA	31.00%	0.0221	41.04%	0.0443	49.71%	0.0391	51.94%	0.0535	59.58%	0.0627
WOA-LFGA	38.29%	0.0004	56.72%	0.0038	72.16%	0.0074	88.75%	0.0010	93.71%	0.0027

**Table 12 biomimetics-08-00354-t012:** Parameters of WSN coverage optimization problem in Section 3.3.

Parameter	Value
Region size	440,400 m^2^
Sensing range	100 m
Sensor nodes number N	13
Individual number	50
Iterations	200
Test times	30

**Table 13 biomimetics-08-00354-t013:** Coverage ratio comparison of WOA-LFGA with other basic algorithms in practical application.

Method	ave	std	C
SMA	11.4011%	0.0159	0.1229
DOA	53.0607%	0.0530	0.5722
AOA	52.3511%	0.0306	0.5645
BWO	52.2743%	0.0579	0.5637
WOA	37.2967%	0.0935	0.4022
WOA-LFGA	83.7718%	0.0035	0.9033

**Table 14 biomimetics-08-00354-t014:** Coverage ratio comparison of WOA-LFGA with different modified WOA in practical application.

Method	ave	std	C
CWOA	53.4095%	0.0666	0.5759
WOABAT	37.4971%	0.0527	0.4043
RDWOA	51.3324%	0.0508	0.5535
WOAmM	38.7471%	0.0452	0.4178
EGE-WOA	43.9632%	0.0366	0.4741
WOA-LFGA	83.7718%	0.0035	0.9033

## Data Availability

Not applicable.

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
