# Peer review of "Application of an Enhanced Whale Optimization Algorithm on Coverage Optimization of Sensor"

_biomimetics, 2023, doi:10.3390/biomimetics8040354_

Round 1
Reviewer 1 Report (Previous Reviewer 3)
The second version of the article contains corrections in accordance with the comments of the reviewers.
I have two comments on the text of the work.
L144 - 2.Related Work. The whole overview is actually in the Introduction. It is necessary either to remove line L144, or to divide the text in the Introduction paragraph into an Introduction and Related Work.
4.2.1 Comparison of WOA-LFGA with other basic algorithms
Authors tested the coverage of 27 target points within an area of 100m×100m. That is, a square area was considered, the algorithm determined the location of the sensors to optimize coverage. In real problems, the area can have various shapes (not necessarily square). Have the authors tested the work of their algorithm on areas with other shapes?
Author Response
Dear Editors and Reviewers:
Thank you for your letter and for the reviewers’ comments concerning our article biomimetics-2525555 entitled "Application of an Enhanced Whale Optimization Algorithm on Coverage Optimization of Sensor", which we submitted to the Biomimetics. Those comments are all valuable and very helpful for revising and improving our paper, as well as the essential guiding significance to our research. We have studied the comments carefully and have made corrections which we hope meet with approval.
In the revised manuscript, we have used green text to indicate the modifications made based on your suggestions. The main corrections in the paper and the responses to the reviewer’s comments are as follows:
Responds to the reviewer’s comments:
Reviewer 1:
- L144 - 2.Related Work. The whole overview is actually in the Introduction. It is necessary either to remove line L144, or to divide the text in the Introduction paragraph into an Introduction and Related Work.
Response: Ok, we have removed L144 - 2.Related Work.
- 4.2.1 Comparison of WOA-LFGA with other basic algorithms
Authors tested the coverage of 27 target points within an area of 100m×100m. That is, a square area was considered, the algorithm determined the location of the sensors to optimize coverage. In real problems, the area can have various shapes (not necessarily square). Have the authors tested the work of their algorithm on areas with other shapes?
Response: It is really true as reviewer suggested that we should test the work on areas with other shapes. We have added relevant experiments. Thank you very much for your suggestion.
We tried our best to improve the manuscript. These changes will not influence the content and framework of the paper. We appreciate for Editors/Reviewers’ warm work earnestly and hope that the correction will meet with approval. Once again, thank you very much for your comments and suggestions.
Reviewer 2 Report (New Reviewer)
The authors have proposed modified Whale Optimization Algorithm combining Levy flight and a genetic algorithm optimization mechanism. The developed algorithm is tested on the set of 35 functions and wireless sensor network coverage optimization problem. The developed algorithm is is shown to be more efficient than the standard literature algorithms. It is original contribution of authors.
The paper is well organized but partially not clearly written. Due to use of the undefined symbols or the lack of suitable explanation the formulation of wireless sensor network coverage optimization problem or the modified whale algorithm are not fully clear. A few figures require either explanation or correction. The paper requires minor corrections before publication. The detailed remarks are attached in the form of pdf file.

English should be verified throughout the paper.
For instance: word Lévy rather than Levy should be used, style of sentence in lines 7 and 8 could be improved, this paper proposes rather than proposed, line 191 "to get the current to the position ....." and many many others.
Author Response
Dear Editors and Reviewers:
Thank you for your letter and for the reviewers’ comments concerning our article biomimetics-2525555 entitled "Application of an Enhanced Whale Optimization Algorithm on Coverage Optimization of Sensor", which we submitted to the Biomimetics. Those comments are all valuable and very helpful for revising and improving our paper, as well as the essential guiding significance to our research. We have studied the comments carefully and have made corrections which we hope meet with approval.
In the revised manuscript, we have used red text to indicate the modifications based on your suggestions. The main corrections in the paper and the responses to the reviewer’s comments are as follows:
Responds to the reviewer’s comments:
Reviewer 2:
- Section 2.1 - formulation of the WSNC optimization problem is not clear. Cost functional C in (5) is maximized or minimized? What are design variables? What are constraints? In line 154 d denotes distance while in lines 153 and 156 symbol d is used? In formula (4) indices i, j should be used rather than symbols x, y denoting spatial variables in line 149?
Response: Cost functional C in (5) is maximized. s1, s2,…,sm are design variables. The constraint is that the sensor nodes must remain within the specified range. We have added an explanation for this constraint in lines 171-174. In line 154 d denotes distance, and in lines 153 and 156 symbol d denotes distance too. represents the distance between si and tj. In formula (4) indices i, j should be used. We have revised formula (4).
- Section 2.2 - three update formulas (5), (7), (13) for X(t+1) are provided. Which one has been used in computations? Correct the notation: the position vector of the whale in the current iteration is denoted by X in line 184 and by X in remaining lines of this section. Definition in lines 199-200 is not clear.
Response: These formulas have been used in calculations. We have provided a detailed explanation of the specific content and utilization of these equations in lines 182-185. Correct the notation: We have corrected the other occurrences of X to X. We have made corrections to lines 199-200, which are now lines 205-206.
- Section 2.3 - formula (14) in line 223 is not clear. For instance: symbol p2 is not defined and do not appear in the right hand side? Is there any relation between formulas (14) and (13), (7), (5)? Line 229 - S denotes in line 149 sensor area? Please write L´evy rather than Levy.
Response: We added a definition for p2 in L224, along with further elaboration on the usage of the formula (14). We provided an explanation of the relationship between equation (14) and the other three equations in L237-238. I apologize for the confusion. In line 229, "s" does not represent the sensor area "S". They are differentiated by their capitalization. Here, "s" represents the step length of the Lévy flight. We have also made corrections throughout the article, replacing all instances of "Levy" with " Lévy ".
- line 254, Figure 1 - The quality of this figure should be improved.
Response: Thank you very much for your suggestion. We have redrawn Figure 1 and exported it as an EMF file format. It has been inserted into the revised manuscript.
- line 276 - Tent mapping is used from literature or it is original work of authors? Insert suitable citation?
Response: Tent mapping is used from other literature, we have added a citation at this location.
- line 349, Algorithm 1: WOA-LFGA - Step 6 requires explanation. The use of updates (7), (12), (14) depends on parameters p and A. In lines 187 and 205 formulas (7) and (12) do not depend on p? Formula (14) in line 223 does not depend neither on p nor A?
Response: It is really true as reviewer suggested that Step 6 requires an explanation. We added an explanation in section 2.2 regarding the dependence of formula usage on A and p.
- line 373, Table 1 - D denotes dimensionality of the problem? What denotes n for function F4?
Response: D denotes the dimensionality of the problem, while "n" also signifies the dimension of the problem. We have rectified the function for F4 and sincerely apologize for the error.
- line 375, Table 2 - functions F22 − F29 - what denotes D for these functions?
Response: Thank you for clarifying. In F22-F29, "D" still represents the dimension of the problem. For example, in F22, the independent variables only include x1 and x2, so the dimension of the problem is 2.
- line 377 - what fitness function is used?
Response: In section 4.1, the fitness function utilizes the functions F1-F35 from Table 1-Table 3.
- line 406 - Fig 3 rather than 1?
Response: We sincerely apologize for the error made during our writing process. We have made the necessary correction.
- line 414, Figure 3, subfigures F1 and F3 - why graphs for WOA-LFGA algorithm end so quickely?
Response: To make the picture more intuitive, the semi-logarithmic graph is used in this paper to reflect the decline rate of the fitness curve, so the part not shown in the graph represents that its fitness value has declined to infinitesimal. The rapid convergence of WOA-LFGA can be attributed to its global search capability. Empirical evidence has shown that WOA-LFGA exhibits strong global search ability, allowing it to find global optimal solutions in a relatively short period of time, while avoiding getting trapped in local optima.
- line 454, 455 - Only C is used as the cost functional? Why WOA-LFGA before modification (23)-(24) has failed to solve this optimization problem? Is there any relation between test results in subsection 4.1 and the results in section 4.2 based on different update formula?
Response: f in Eq. (4) and C in Eq. (5) are used as the objective function when solving the WSN optimization model. The WOA-LFGA also optimize this problem even before being modified. However, multiple experimental validations have shown significant performance improvements after a simple modification based on equation (23). Therefore, we have included this modified equation in the experiments of this section. In Section 4.1, we demonstrated the universality of WOA-LFGA through experiments. In Section 4.2, we made minor adjustments specific to the problem, which further highlights the strong competitiveness of WOA-LFGA in optimization experiments.
- lines 504-508 - how strongly these results depend on suitable choice of the update (23)?
Response: Without equation (23), the WOA-LFGA had only a slight advantage of approximately 2% in terms of coverage rate compared to the second-place method. However, with the inclusion of equation (23), the WOA-LFGA's coverage rate surpasses the second-place form by over 10%. This method demonstrates the significance of equation (23) in the optimization process.
- line 520 - Prim algorithm is cited from literature or it is original work of authors?
Response: Prim algorithm is cited from the literature; we added a citation to the algorithm in the manuscript.
We tried our best to improve the manuscrip. These changes will not influence the content and framework of the paper. We appreciate for Editors/Reviewers’ warm work earnestly and hope that the correction will meet with approval. Once again, thank you very much for your comments and suggestions.
This manuscript is a resubmission of an earlier submission. The following is a list of the peer review reports and author responses from that submission.
Round 1
Reviewer 1 Report
The topic is interesting and practical since IoT is a hot trend nowadays, and the problem of low coverage is a serious issue in this topic. However, there are the following major issues in this manuscript that should be addressed before publishing.
1. It is suggested to mention the novelty and findings of this study in the abstract.
2. The importance of this study is unclear. It is suggested to boost the literature review and review the WOA variants applying in different application such as Enhanced whale optimization algorithm for medical feature selection: A COVID-19 case study and Hybridizing of whale and moth-flame optimization algorithms to solve diverse scales of optimal power flow problem. It is recommended to see A Systematic Review of the Whale Optimization Algorithm: Theoretical Foundation, Improvements, and Hybridizations.
3. The contribution of this study should be summarized at the end of the Introduction.
4. Equation 18, the operator should be defined before levy(s).
5. It is suggested to visualize a curve plot for Equation 20.
6. The title of Algorithm 1 should translate into English. Please check the whole of the manuscript for it.
7. Step 6 in algorithm 1 is unclear.
8. The recent CEC test functin is sugetesed.
9. It is suggested to compare the proposed algorithm with state-of-the-art optimizers such as QANA.
10. The quality of Figure 1 should be improved.
11. It is unclear why the different contender algorithms were used to assess the ability of the proposed algorithm in the CEC test function and WSN problem. It is suggested to consider a similar set of contender algorithms for two groups of problems.
12. The success rate should be computed.
Author Response
- It is suggested to mention the novelty and findings of this study in the abstract.
Response: We have modified the abstract section and mentioned the novelty and findings of this study.
- The importance of this study needs to be clarified. It is suggested to boost the literature review and review the WOA variants applied in different applications such as Enhanced whale optimization algorithm for medical feature selection: A COVID-19 case study and hybridizing of whale and moth-flame optimization algorithms to solve diverse scales of optimal power flow problem. It is recommended to see A Systematic Review of the Whale Optimization Algorithm: Theoretical Foundation, Improvements, and Hybridizations.
Response: Thank you very much for your suggestion. We have strengthened our literature review. We reviewed the variants of WOA used in different applications and the hybrid algorithms of WOA with other algorithms (L91-L123).
- The contribution of this study should be summarized at the end of the Introduction.
Response: OK. We have added it at the end of the Introduction. (L136-L143)
- Equation 18, the operator should be defined before levy(s).
Response: Thank you for your prompt. We have defined Eq. (18) before levy(s). (L219)
- It is suggested to visualize a curve plot for Equation 20.
Response: We have visualized a curve plot for Equation 20 (figure 2). (L318)
- The title of Algorithm 1 should translate into English. Please check the whole of the manuscript for it.
Response: We are very sorry for our negligence of this careless writing in this paper. We have made modifications to the title of Algorithm 1, and we have also corrected other errors.
- Step 6 in algorithm 1 needs to be clarified.
Response: Thank you for pointing out this issue. We have provided a more detailed description of step 6.
- The recent CEC test function is suggested.
Response: We are sorry we cannot complete this new testing experiment quickly. We temporarily used the previous test set because many authors used these CEC test functions in many other works of literature. Another reason is that the test functions used in this paper, although not recently proposed, are still very classic and can demonstrate the effectiveness and superiority of the algorithm. But as an additional effort, we increased the dimensions of some test functions to verify the algorithm’s efficacy.
- It is suggested to compare the proposed algorithm with state-of-the-art optimizers such as QANA.
Response: We are sorry we haven't had enough time to modify this suggestion. The comparison algorithms used in our study are also very typical and commonly used, many of which have been proposed in the last five years.
- The quality of Figure 1 should be improved.
Response: The original Figure 1 in the manuscript is now Figure 3 in this manuscript. We have redrawn these figures and saved them in EMF file format. This format can maintain the accuracy of the figures and still display clearly when the figures are enlarged.
- It is unclear why the different contender algorithms were used to assess the ability of the proposed algorithm in the CEC test function and WSN problem. It is suggested to consider a similar set of contender algorithms for two issue groups.
Response: We are very sorry that, as time constraints limit the potential for significant modifications to the experimental section. In the optimization experiment for functions, we used 5 standard algorithms for comparison to highlight the universality of WOA-LFGA. In the experiment of WSN coverage optimization, we have selected several new algorithms proposed in recent years and several improved algorithms based on WOA to compare and highlight the superiority of WOA in practical applications. The use of different competitor algorithms for evaluation is to highlight the competitiveness of our algorithm from different perspectives.
- The success rate should be computed.
Response: Sorry, we cannot understand which success rate you are referring to. Could you specifically refer to a specific section?

Reviewer 2 Report
1. Try to improve the format of mathematical symbols. There are a lot of math symbols, but they are not in the correct form outside the displayed formulas.
2. I think you should put some words to explain the "dot notation" in equations (5) to (9) (L131 to L130). It looks like the "inner product", but it cannot be because you treat the results as vectors.
3. The title of section 3 (L189) is totally wrong. I believe you plant the wrong words.
4. There are many careless writing in this paper. For example, L159 (S & s), L191 (WOa), L233 (\alpha zero), L275 (Chinese sentence appears), L393 (Lfga). Please correct them.
5. In my opinion, I believe the WSN problem proposed in this paper is easy to solve: you can simply deploy the sensors uniformly in the region. Figure 4(f) is the best result made by your proposed algorithm, and it also supports my observation. I suggest the authors try to make the test problem harder, in order to demonstrate the advantages of the proposed method.
6. (L303) The population size is only 30. This is really a small number. Try to make it bigger.
The introduction is carefully written. However, this is not the case after Section 2. For example, there are two "can"s in L223 & 224, two verbs in L268.
Author Response
- Try to improve the format of mathematical symbols. There are many math symbols, but they need to be in the correct form outside the displayed formulas.
Response: Thank you very much for your reminder. We have corrected mathematical symbols outside the displayed formulas.
- I think you should put some words to explain the "dot notation" in equations (5) to (9) (L131 to L130). It looks like the "inner product", but it cannot be because you treat the results as vectors.
Response: It is true as the reviewer said, the "dot notation" is not the "inner product,” and we have added an explanation for it behind equations (9). (L190-L191)
- The title of section 3 (L189) is totally wrong. I believe you plant the wrong words.
Response: We are very sorry for our incorrect writing. We have made corrections to the title of section 3.
- There is much careless writings in this paper. For example, L159 (S & s), L191 (WOa), L233 (\alpha zero), L275 (Chinese sentence appears), L393 (Lfga). Please correct them.
Response: We are very sorry for our negligence in this sloppy writing in this paper. We have made modifications to these parts of the reviewer expenses, and we have also corrected other errors.
- In my opinion, I believe the WSN problem proposed in this paper is easy to solve: you can deploy the sensors uniformly in the region. Figure 4(f) is the best result made by your proposed algorithm and supports my observation. I suggest the authors try to make the test problem harder, demonstrating the proposed method's advantages.
Response: Thank you very much for your advice. We have revised the experimental section in Section 4.2 by introducing coverage efficiency to evaluate node redundancy and increase the workload. As such, we conducted more experiments to verify the algorithm’s effectiveness.
- (L303) The population size is only 30. This is a small number. Try to make it bigger.
Response: Thank you very much for your suggestion. We have added the experiment section in this part. We increased the value of d to 50, 100, and 500, respectively.
7.The introduction is carefully written. However, this is different after Section 2. For example, there are two "can"s in L223 & 224, two verbs in L268.
Response: Thank you very much for your reminder. We have corrected these two errors. At the same time, we also checked the introduction.

Reviewer 3 Report
The manuscript is devoted to the actual topic - ensuring coverage of a given territory by sensors in the WSN. This paper proposes a new advanced whale optimization algorithm (WOA) incorporating Levy flight and a Genetic Algorithm Optimization Mechanism (WOA-LFGA). The simulation results demonstrate the high competitiveness of the improved algorithm compared to conventional algorithms.
Unfortunately, the authors did not describe their study well enough. The text lacks the detailed description of the methods used, and the formulas used are not accurately described. Below are a few notes.
Abstract .
“29 mathematical optimization problems and a WSN coverage optimization model evaluated with WOA-LFGA”. The authors need to write this sentence more correctly.
1. Introduction
The Introduction should contain a general description of the problem in WSN – the providing sufficient coverage of the territory by sensors. “The wireless sensor coverage optimization problem is solvable with optimization strategies. Recently, numerous researchers have proposed different swarm intelligent optimization algorithms (SIA) to tackle such problems”. It seems to me, that it is necessary to describe what other methods this problem have been solved, what results were obtained in various studies.
2. Related Work
The text in this section consists only of this sentence: “Section 2.1 describes the wireless sensor network coverage model. Sections 2.2-2.4 93 introduce the basic concepts and formulas of relevant algorithms”. Related work have been described in Introduction. Why? Authors need to pay more attention to the structure of the article.
2.1 Wireless Sensor Network Coverage Model
Authors made a mistake in writing formula (2) - the Euclidean distance between the sensor node and the monitored node. Check this formula, please. The formula (4): What does it mean: L×W?
2.2 Overview of Whale Optimization Algorithm (WOA)
The authors should make the text in this section more understandable. Not all variables have descriptions.
2.3 The Lévy Flights’ Method
In this section, the description of Lévy method should be made more clearer. Authors should provide references to some articles describing the Lévy Flights’ Method.
Variables and Abbreviations.
The text of manuscript uses many variables, formulas. Authors need to describe in more detail all the formulas and the variables included in them. There are many abbreviations in section 4.2.1 (409-413). Some of these abbreviations have been decoded, for others I could not find a decryption.

Author Response
Abstract.
“29 mathematical optimization problems and a WSN coverage optimization model evaluated with WOA-LFGA”. The authors need to write this sentence more correctly.
Response: We have made corrections according to the Reviewer’s comments. We have rewritten this sentence.
- Introduction
The Introduction should contain a general description of the problem in WSN – the providing sufficient coverage of the territory by sensors. “The wireless sensor coverage optimization problem is solvable with optimization strategies. Recently, numerous researchers have proposed different swarm intelligent optimization algorithms (SIA) to tackle such problems”. It seems to me, that it is necessary to describe what other methods this problem have been solved, and what results were obtained in various studies.
Response: Thank you very much for your suggestion. We have added a summary of other methods in the second paragraph of the introduction. (L35-L53)
- Related Work
The text in this section consists only of this sentence: “Section 2.1 describes the wireless sensor network coverage model. Sections 2.2-2.4 93 introduce the basic concepts and formulas of relevant algorithms”. Related work has been described in Introduction. Why? Authors need to pay more attention to the structure of the article.
Response: It is accurate as Reviewer suggested that related work has been described in Introduction. So, we deleted the introduction in the related work.
2.1 Wireless Sensor Network Coverage Model
The authors made a mistake in writing formula (2) - the Euclidean distance between the sensor node and the monitored node. Check this formula, please. The formula (4): What does it mean: L×W?
Response: We are very sorry for our incorrect writing. We have rewritten Formula 2. We also added an explanation for L×W before formula 4. (L160-L161)
2.2 Overview of Whale Optimization Algorithm (WOA)
The authors should make the text in this section more understandable. Not all variables have descriptions.
Response: Thank you very much for your prompt. We have provided a more detailed description of this section.
2.3 The Lévy Flights’ Method
In this section, the description of Lévy method should be made more clearer. Authors should provide references to some articles describing the Lévy Flights’ Method.
Response: Thank you very much for your suggestion. We have added the reference [20] that describes the Lévy Flights’ Method in the article. (L217)
Variables and Abbreviations.
The text of manuscript uses many variables, formulas. Authors need to describe in more detail all the formulas and the variables included in them. There are many abbreviations in section 4.2.1 (409-413). Some of these abbreviations have been decoded, for others I could not find a decryption.
Response: It is accurate as Reviewer suggested that we need to describe all the formulas and variables in more detail. We are very sorry for the errors made during the writing process regarding the abbreviations in section 4.2.1. Those abbreviations do not belong to this section. They are the content of section 4.2.2. We have removed these abbreviations in section 4.2.1.

Round 2
Reviewer 1 Report
There are still some critical concerns about the effectiveness of the proposed WOA-LFGA that the authors do not effectively address. Proposing a new variant for WOA is not enough; the authors should prove why the new WOA variant is presented in this study. This question can be satisfied by several experiments on the algorithm's effectiveness and by comparing it with WOA variants and well-known competitor algorithms to determine if the proposed WOA-LFGA can overcome these algorithms and obtain the first rank in effectiveness. Moreover, many WOA variants improved with the Lévy flight, like WOA-LFGA. Therefore there are many concerns about the repetition and novelty of this paper. The other comments are summarized as follows.
1. It is suggested to provide experimental evaluation and show how the Lévy flight helps WOA-LFGA overcome the WOA weaknesses.
2. There are the same sentences in the Abstract. It should rewrite the Abstract with informative sentences showing the challenge, necessity, novelty, and finding. Researchers need to know why WOA improved with Levy flight again.
3. The description of the benchmark function can be removed or added in the appendix.
4. It is suggested to compare the proposed algorithm with state-of-the-art optimizers such as QANA and recent WOA variants.
5. It is unclear why the following comments are not responded. These comments can improve the quality of the paper and satisfy why the new WOA variants with the same proposed strategy as the existing WOA variants are presented.
C1. The recent CEC test function is suggested. “Response: We are sorry we cannot complete this new testing experiment quickly. We temporarily used the previous test set because many authors used these CEC test functions in many other works of literature. Another reason is that the test functions used in this paper, although not recently proposed, are still very classic and can demonstrate the effectiveness and superiority of the algorithm. But as an additional effort, we increased the dimensions of some test functions to verify the algorithm’s efficacy.”
C2. It is unclear why the different contender algorithms were used to assess the ability of the proposed algorithm in the CEC test function and WSN problem. Considering a similar set of contender algorithms for two issue groups is suggested. “Response: We are very sorry that, as time constraints limit the potential for significant modifications to the experimental section. In the optimization experiment for functions, we used 5 standard algorithms for comparison to highlight the universality of WOA-LFGA. In the experiment of WSN coverage optimization, we have selected several new algorithms proposed in recent years and several improved algorithms based on WOA to compare and highlight the superiority of WOA in practical applications. The use of different competitor algorithms for evaluation is to highlight the competitiveness of our algorithm from different perspectives. “
C3. The success rate should be computed. “Response: Sorry, we cannot understand which success rate you are referring to. Could you specifically refer to a specific section?”
C4. The formula can be found in QANA or by skimming other papers.